# Gifts from anomalies:
# Exact results for Landau phase transitions in metals

Zhengyan Darius Shi[1], Hart Goldman[1,*], Dominic V. Else[2,1,*], and T. Senthil[1]

[1]*Department of Physics, Massachusetts Institute of Technology, Cambridge, MA 02139, USA*

[2]*Department of Physics, Harvard University, Cambridge, MA 02138, USA*

May 26, 2022

## Abstract

Non-Fermi liquid phenomena arise naturally near critical points of Landau ordering transitions in metallic systems, where strong fluctuations of a bosonic order parameter destroy coherent quasiparticles. Despite progress in developing controlled perturbative techniques, much of the low energy physics of such metallic quantum critical points remains poorly understood. We demonstrate that *exact, non-perburbative* results can be obtained for both optical transport and static susceptibilities in "Hertz-Millis" theories of Fermi surfaces coupled to critical bosons. Such models possess a large emergent symmetry and anomaly structure, which we leverage to fix these quantities. In particular, we show that in the infrared limit, the boson self energy at zero wave vector, $\mathbf{q} = 0$, is a constant independent of frequency, and the real part of the optical conductivity, $\sigma(\omega)$, is purely a delta function Drude peak with no other corrections. Therefore, further frequency dependence in the boson self energy or optical conductivity can only come from irrelevant operators in a clean system. Exact relations between Fermi liquid parameters as the critical point is approached from the disordered phase are also obtained. The absence of a universal, power law frequency dependence in the boson self energy contrasts with previous perturbative calculations, and we explain the origin of this difference.

---

* These authors contributed equally to the development of this work.

# 1 Introduction

The problem of understanding metallic phases without stable quasiparticles, broadly dubbed non-Fermi liquids (NFLs), continues to be a major enduring challenge. Despite the tremendous successes of Landau's Fermi liquid theory in describing conventional metals, many systems in nature appear to host metallic states with properties that are inconsistent with its quasiparticle paradigm. Examples include the strange metal phases arising in high temperature superconductors [1–3], heavy fermion materials [4, 5], iron pnictides [6], and a growing number of other materials [7–10]; "anomalous metal" phases appearing in superconducting thin films [11]; and the metallic states at even denominator filling in quantum Hall systems [12, 13]. Because such systems cannot be adequately described by a Fermi surface with stable quasiparticle excitations, strong interactions must play an essential role.

A natural situation where NFL physics is expected arises when a metal is near a quantum critical point (QCP). In the simplest scenarios, the metallic degrees of freedom coexist at the QCP with critical fluctuations of an order parameter, which in turn mediates scattering of quasiparticles at all energy scales. For this reason, considerable attention has been devoted to a simple class of effective field theories in which the low energy degrees of freedom near a Fermi surface are coupled to a critically fluctuating boson. This framework for constructing theories of metallic quantum critical points was developed by Hertz and Millis [14, 15] and has been adapted for a wide class of examples [4], including nematic orders [16, 17], antiferromagnetic order [18–22], and Fermi surfaces coupled to fluctuating gauge fields [1, 23–27]. In three spatial dimensions, the physics of this model is believed to be well described through the original approach of Hertz and Millis, who constructed an effective theory for the order parameter by integrating out the entire Fermi surface of gapless fermion degrees of freedom (an approach to studying the three dimensional theory with the Fermi surface still present was developed in Ref. [28]). However, many salient experimental observations of metallic quantum critical phenomena occur in two dimensional systems, where the Hertz-Millis approach breaks down.

In two dimensions, much of the low energy physics of Fermi surfaces coupled to critical bosons remains difficult to access with any theoretical control. The simplest perturbative approach using a large-$N$ expansion in the number of fermion species has been demonstrated to suffer from infrared (IR) issues, rendering it uncontrolled [29]. Other attempts at controlled perturbative deformations involve spatial non-locality [26, 30, 31], varying the co-dimension of the Fermi surface [32], introducing matrix bosons [28, 33, 34], and, most recently, adding random couplings between different species of bosons and fermions [35, 36]. While each of

these approaches may allow access to a formally controlled IR fixed point the connection of such a fixed point to the original problem is not always clear. Moreover, a systematic understanding of some observable features of these models, such as transport, is generally lacking. It is therefore of great importance to develop general constraints on theories of Fermi surfaces coupled to critical bosons that do not rely on perturbative deformations.

In this work, we consider the important class of theories where the bosonic field has gapless fluctuations near zero momentum. We show that several quantities in these theories can be extracted through *exact, non-perturbative* arguments. In particular, our main focus will be on optical properties; namely, the boson propagator, $D(\omega)$, and the optical conductivity, $\sigma(\omega)$, both evaluated at wave vector $\mathbf{q} = 0$. Our only assumptions are that (1) certain terms in the microscopic action will be irrelevant in the IR and can be discarded, such that we may consider an effective theory in which the Fermi surface is decomposed into small patches, and (2) that this "mid-IR" patch theory flows to a stable IR fixed point describing a second-order quantum phase transition. With these assumptions, we find that at low energies,

$$D(\omega) = \frac{1}{\Pi_0} \,, \qquad \sigma(\omega) = \frac{\mathcal{D}(\Pi_0)}{\pi} \frac{i}{\omega} \,. \tag{1.1}$$

where $\Pi_0$ is a constant independent of frequency and $\mathcal{D}(\Pi_0)$ is another constant depending on $\Pi_0$, the Fermi velocity, and the symmetry properties of the boson-fermion coupling. Both $\Pi_0$ and $\mathcal{D}(\Pi_0)$ can be determined exactly from our general arguments. An important consequence of these results is that at the IR fixed point, neither quantity can exhibit universal scaling behavior in frequency: the conductivity is a Drude peak alone. Critical fluctuations do not generate any additional frequency-dependent conductivity in the IR. Thus, in the clean limit, the only way frequency scaling can arise is through irrelevant operators. The effects of various irrelevant operators have been explored in Refs. [27, 37–39] within the Hertz-Millis framework.

Furthermore, away from criticality in the disordered phase, the NFL theory flows to a Fermi liquid fixed point characterized by charge densities, $\widetilde{n}_\theta$, at angle $\theta$ on the Fermi surface. Our non-perturbative arguments also enable us to obtain exact expressions for static density susceptibilities, $\chi_{\widetilde{n}_\theta \widetilde{n}_\theta}$, of the Fermi liquid as it approaches the quantum critical point. Such results allow us to demonstrate precisely how these susceptibilities (and the associated Landau parameters) of the Fermi liquid diverge as the quantum critical point is approached from the Fermi liquid regime. Our results for these susceptibilities are in agreement with expectations in the existing literature [30, 40] but significantly clarify the theoretical structure that controls their singularities. We also determine the divergence of the order parameter susceptibility upon approaching criticality from the Fermi liquid regime,

finding agreement with the result stated in Ref. [17] through a somewhat different argument.

Our exact approach is based on the following pair of observations. First, the models we consider host a very large emergent symmetry group associated with conservation of charge at every point on the Fermi surface [41], since long wavelength fluctuations of the boson only lead to forward scattering. Second, the dispersion at every point on the Fermi surface is chiral, leading to an "anomaly." The presence of the anomaly means that the emergent charge conservation symmetry at each point on the Fermi surface, which is present at the classical level, is deformed in a precise, quantized way in the quantum theory. Both of these observations are non-perturbative features of the model. In this work, we show that, together with the operator equations of motion for the boson field, they place enough constraints on the physics to derive optical transport, Eq. (1.1), and static susceptibilities.

The large emergent symmetry and anomaly structure of the critical NFL metal is in fact related to the emergent symmetry [42–47] and corresponding anomaly [41] of an ordinary Landau Fermi liquid metal. In particular, this class of NFL metals provides an example of the "ersatz Fermi liquids" introduced in Ref. [41], a concept which has been explored further in a number of recent works [48–53]. We emphasize, however, that the contraints we derive pertain only to the particular class, of models considered in this paper and not necessarily to a general ersatz Fermi liquid.

We proceed as follows. In Section 2, we introduce the main flavor of our arguments by considering how the axial anomaly fixes the features of a simple, exactly solvable 1D model of a Fermi surface coupled to a fluctuating gauge field, known as the Schwinger model. In Section 3, we describe the 2D model of Fermi surface coupled critical boson that is the main focus of this work, and we present our main results. In particular, we show how anomaly arguments can fix the boson propagator at $q = 0$, the optical conductivity, and the static density susceptibilities in the IR limit. In Section 4, we consider how our symmetry and anomaly ideas can allow us to determine how different classes of irrelevant operators can affect these conclusions. In Section 5, we discuss how our results relate to earlier perturbative calculations [27, 37, 54, 55] appearing to yield non-trivial frequency scaling of the conductivity or boson self energy in the IR limit, in apparent contrast to Eq. (1.1). In Section 6, we discuss various deformations of the model that have been proposed to yield controlled perturbative expansions, and we consider to what extent our arguments apply to these models. We conclude in Section 7. Three Appendices provide some technical details.

# 2 Warm-up: the axial anomaly and transport in 1D

## 2.1 The free Dirac fermion

Before studying theories of metallic quantum critical points, we review the extensive role played by the axial anomaly in constraining the physics of fermions in one spatial dimension. Consider first the theory of a free Dirac fermion with charge $e = 1$, $\psi = (\psi_+, \psi_-)$, coupled to a background gauge field, $A_\mu$,

$$S = \int dt dx \, \bar{\psi} (i\partial_\mu - A_\mu) \gamma^\mu \psi \tag{2.1}$$

where $\gamma^\mu = (\sigma^y, i\sigma^x)$, $\{\gamma^\mu, \gamma^\nu\} = 2\eta^{\mu\nu}$, $\bar{\psi} = \psi^\dagger \gamma^0$, and we work with the metric signature, $(+, -)$. We also define $\gamma_\star = \gamma^0 \gamma^1 = \sigma^z$, which anticommutes with $\gamma^\mu$. Physically, the components of $\psi$ correspond respectively to fermions moving to the left ($\psi_+$) and right ($\psi_-$) with velocity, $v = 1$. This theory may be interpreted as the low energy limit of non-relativistic fermions at finite density in 1D, with $\psi_\pm$ corresponding to excitations near the two Fermi points.

At the classical level, this theory has two global U(1) symmetries which rotate $\psi_+$ and $\psi_-$ independently,

$$\mathrm{U}(1)_L : \psi_+ \to e^{i\theta_L} \psi_+, \ \mathrm{U}(1)_R : \psi_- \to e^{i\theta_R} \psi_- \,, \tag{2.2}$$

which would naïvely indicate that the charges of left and right-moving fermions, $Q_\pm = \int dx \, \psi_\pm^\dagger \psi_\pm$, are independently conserved. Moreover, an important property of the $\mathrm{U}(1)_{L,R}$ symmetries in 1D is that the charge density and current are proportional to one another, $J_\pm^\mu = (n_\pm, J_\pm) = (\psi_\pm^\dagger \psi_\pm, \pm \psi_\pm^\dagger \psi_\pm)$.

The $\mathrm{U}(1)_L$ and $\mathrm{U}(1)_R$ symmetries may be re-expressed as linear combinations of so-called vector and axial rotations, which act as

$$\mathrm{U}(1)_V : \psi \to e^{i\alpha_V} \psi \,, \ \mathrm{U}(1)_A : \psi \to e^{i\alpha_A \gamma_\star} \psi = (e^{i\alpha} \psi_+, e^{-i\alpha} \psi_-) \,. \tag{2.3}$$

Here $\mathrm{U}(1)_V$ is the physical electromagnetic (EM) symmetry, with current, $J^\mu = (\rho, J^x) = \bar{\psi} \gamma^\mu \psi$, and $\mathrm{U}(1)_A$ is the axial symmetry, which acts oppositely on the left and right-moving fermions and has current, $J_\star^\mu = \bar{\psi} \gamma^\mu \gamma_\star \psi$.

An important feature of this model is the so-called *chiral anomaly*, according to which the charges of left and right-movers are not independently conserved in the presence of a background gauge field for the EM charge (or vice versa),

$$\partial_\mu J_+^\mu = -\frac{1}{2\pi} E \,, \ \partial_\mu J_-^\mu = \frac{1}{2\pi} E \,, \tag{2.4}$$

where $E = -F_{tx} = -\partial_t A_x + \partial_x A_t$ is the electric field. This implies that the physical EM current, $J_V^\mu$, continues to be conserved, while the axial current is not,

$$\partial_\mu J_\star^\mu = -\frac{1}{\pi} E \,, \tag{2.5}$$

Eq. (2.5) is referred to as the axial anomaly. Such violations of global symmetries due to the introduction of external fields are known as an 't Hooft anomaly between the vector and axial U(1) symmetries. Anomalies are discrete topological properties of field theories; for example, one can show[1] that the coefficient of the right-hand side of Eq. (2.5) is quantized due to charge quantization.

An additional property of the 1D problem is that the physical charge density is the axial current and vice versa,

$$J_\star^\mu = (\psi_+^\dagger \psi_+ - \psi_-^\dagger \psi_-, \psi_+^\dagger \psi_+ + \psi_-^\dagger \psi_-) = (J^x, \rho) = -\varepsilon^{\mu\nu} J_\nu \,. \tag{2.6}$$

One can intuitively see how this is related to the anomaly. Indeed, establishing an electric field by coupling to the gauge field, $A_\mu$, will lead to a net current of left or right-moving fermions. But because the physical current is the axial charge, the axial charge cannot possibly be conserved.

From the anomaly equation, Eq. (2.5), combined with Eq. (2.6) and charge conservation, $\partial_\mu J^\mu = \partial_t \rho + \partial_x J^x = 0$, we can immediately read off the response of the free Dirac theory to electric fields. For example, we can obtain the conductivity by partial differentiating both sides of the anomaly equation,

$$(\partial_t^2 - \partial_x^2)J^x = -\frac{1}{\pi}\partial_t E \,, \tag{2.7}$$

which in frequency and momentum space[2] implies

$$J_x(\omega, q) = \frac{i}{\omega}\left(\frac{1}{\pi}\frac{\omega^2}{\omega^2 - q^2}\right) E(\omega, q) \,, \tag{2.8}$$

This is rather unsurprising: in this example, the derivation of the anomaly ultimately amounts to computing the EM conductivity. Notice that, in the limit $q \to 0$, the optical conductivity takes the Drude form, with weight fixed by the anomaly,

$$\sigma_{xx}(\omega) = \frac{1}{\pi}\frac{i}{\omega} \,. \tag{2.9}$$

---

[1]For a pedagogical review, see Ref. [56].

[2]We use a Fourier transform convention where $f(x^\mu) = \int \frac{d^2q}{(2\pi)^2}e^{-iq_\mu x^\mu}f(q_\mu)$ where $(q_0, q_1) = (\omega, q)$ and $(x^0, x^1) = (t, x)$.

In addition to fixing the gauge invariant observables, the axial anomaly also underlies bosonization, in which Eq. (2.8) is a direct consequence of the bosonization dictionary, $J^\mu = \varepsilon^{\mu\nu}\partial_\nu\varphi$. See, for example, Ref. [57] for a review.

## 2.2 Gauge fluctuations and the Schwinger model

We now review a less trivial example that we will see is intimately related to the problem of a Fermi surface coupled to a critical boson. We again consider a theory of Dirac fermions, $\psi = (\psi_+, \psi_-)$, but now we promote the probe field, $A_\mu$, to a fluctuating degree of freedom, which we will denote $a_\mu$,

$$S = \int dt dx \left[ \bar{\psi}(i\partial_\mu - a_\mu)\gamma^\mu\psi + \frac{1}{2g^2}\,e^2 \right]\,, \tag{2.10}$$

where $e = -f_{tx} = -\partial_t a_x + \partial_x a_t$ is the fluctuating electric field. This theory is commonly referred to as the Schwinger model [58]. Because gauge fluctuations mediate a linear potential in 1D, the Schwinger model is a paradigmatic example of confinement. However, the existence of the axial anomaly allows the Schwinger model to be solved exactly, for example via bosonization [59–61]. Using the anomaly, it is possible to solve for the gauge field propagator exactly, and in the process show that the spectrum contains only a gapped collective excitation that is familiar to condensed matter physicists as a plasmon.

As in the free theory, the Schwinger model classically has vector and axial U(1) symmetries, with currents $j^\mu = \bar{\psi}\gamma^\mu\psi = (\rho, j^x)$ and $j^\mu_\star = \bar{\psi}\gamma^\mu\gamma_\star\psi = (j^x, \rho)$ respectively, only now the vector symmetry has been gauged. The axial anomaly in the Schwinger model is then simply the gauged version of that in the free theory,

$$\partial_\mu j^\mu_\star = \partial_t j^x + \partial_x\rho = -\frac{1}{\pi}\,e\,, \tag{2.11}$$

Note that, up to contact terms, the anomaly may be viewed as an operator equivalence in the quantum theory.

The anomaly encodes the linear response of the fermions to the emergent electric field. To see this, we again differentiate both sides of Eq. (2.11) and invoke physical charge conservation, $\partial_\mu j^\mu = 0$,

$$(\partial_t^2 - \partial_x^2)j^x = -\frac{1}{\pi}\partial_t e\,,\,(\partial_t^2 - \partial_x^2)\rho = \frac{1}{\pi}\partial_x e, \tag{2.12}$$

which upon Fourier transform can be solved to yield the gauge field self energy tensor, $\Pi^{\mu\nu}$,

$$j^\mu(\omega, q) = \Pi^{\mu\nu}(\omega, q)\,a_\nu(\omega, q)\,,\,\Pi^{\mu\nu}(\omega, q) = -\,\Pi(\omega, q)\left(\eta^{\mu\nu} - \frac{q^\mu q^\nu}{\omega^2 - q^2}\right)\,, \tag{2.13}$$

where the tensor structure is fixed by current conservation, $\partial_\mu j^\mu = 0$. One then finds that in the Schwinger model,

$$\Pi(\omega, q) = \frac{1}{\pi}. \tag{2.14}$$

This is a remarkable consequence of the anomaly: despite strong interactions, the anomaly non-perturbatively fixes the fermion response to emergent electric fields.

Moreover, knowing $\Pi(\omega, q)$, one can compute the gauge field propagator. Let us introduce a background source, $h^\mu(t, x)$, for the gauge field,

$$S_{\text{source}} = -\int dt dx \, h^\mu a_\mu. \tag{2.15}$$

The equation of motion for $a_\mu$ in the presence of this source is

$$\frac{1}{g^2} \partial_\nu f^{\nu\mu} - j^\mu = h^\mu. \tag{2.16}$$

We note that this may be interpreted as an operator equation (up to contact terms), as can be observed by adding a probe for the current, $A_\mu j^\mu$ and shifting $a_\mu$ by $A_\mu$. Passing to momentum space and substituting $j^\mu = \Pi^{\mu\nu} a_\nu$, this equation becomes

$$\left[ (D_0^{-1})^{\mu\nu} - \Pi^{\mu\nu} \right] a_\nu(\omega, q) = h^\mu(\omega, q), \tag{2.17}$$

where

$$(D_0^{-1})^{\mu\nu} = -\frac{1}{g^2}[(\omega^2 - q^2)\eta^{\mu\nu} - q^\mu q^\nu] \tag{2.18}$$

is the inverse of the bare Maxwell propagator. On fixing, say, to Lorentz gauge, $\partial_\mu a^\mu = 0$, linear response theory then tells us that the propagator of $a_\mu$ can be read off as

$$D_{\mu\nu}(\omega, q) = i\langle a_\mu(-\omega, -q)a_\nu(\omega, q)\rangle = \left[ (D_0^{-1})^{\mu\nu} - \Pi^{\mu\nu} \right]^{-1} \tag{2.19}$$

$$= -\frac{g^2}{\omega^2 - q^2 - g^2/\pi} \left( \eta_{\mu\nu} - \frac{q_\mu q_\nu}{\omega^2 - q^2} \right). \tag{2.20}$$

The propagator has a pole at $\omega_P^2 = g^2/\pi$, meaning that the gauge field becomes massive! This indicates the presence of a "plasmon" in the UV, which oscillates at plasma frequency $\omega_P$. Note also that, in the infrared (IR) limit[3], $g^2 \to \infty$, the gauge field propagator becomes simply

$$\langle a_\mu(-\omega, -q)a_\nu(\omega, q)\rangle_{g^2 \to \infty} = i(\Pi^{-1})_{\mu\nu}. \tag{2.21}$$

---

[3]We call $g^2 \to \infty$ the IR limit because $g$ is the only mass scale present in the problem.

In terms of Feynman diagrams, $\Pi_{\mu\nu}$ can be understood as the sum of one-particle irreducible (1PI) contributions to the gauge propagator. Indeed, in the limit $g^2 \to \infty$ ($D_0^{-1} \to 0$), the boson propagator is exactly given by the sum of 1PI diagrams.

Using this result, one can compute the response to a background electric field[4] by introducing a probe coupling as $A_\mu j^\mu$. By shifting $a_\mu$ by $A_\mu$, the current-current correlator, $K_{\mu\nu} = i\langle[j_\mu, j_\nu]\rangle$ (we leave commutators implicit throughout the manuscript) can be expressed in terms of the gauge propagator,

$$K_{\mu\nu}(\omega, \boldsymbol{q}) = -i\frac{\delta}{\delta A^\mu}\frac{\delta}{\delta A^\nu}\log Z[A] = -(D_0^{-1})_{\mu\nu} + (D_0^{-1})_{\mu\lambda}D^{\lambda\sigma}(D_0^{-1})_{\sigma\nu} \tag{2.22}$$

$$= [\Pi (1 - \Pi D_0)^{-1}]_{\mu\nu} \tag{2.23}$$

$$= -\frac{1}{\pi}\frac{1}{\omega^2 - q^2 - g^2/\pi}\left[(\omega^2 - q^2)\eta_{\mu\nu} - q_\mu q_\nu\right] . \tag{2.24}$$

Again, we observe a pole at the plasmon frequency. In the Minkowski signature $(+, -)$, the conductivity is related to $K_{\mu\nu}$ as

$$j_x(\omega, \boldsymbol{q} = 0) = K_{xx}(\omega, \boldsymbol{q} = 0)a^x(\omega, \boldsymbol{q} = 0) = \frac{i}{\omega}K_{xx}(\omega, \boldsymbol{q} = 0)E_x(\omega, \boldsymbol{q} = 0) . \tag{2.25}$$

Thus, as in the free Dirac case, the anomaly has allowed us to determine the optical conductivity,

$$\sigma_{xx}(\omega) = \frac{i}{\omega}K_{xx}(\omega, \boldsymbol{q} = 0) = \frac{i}{\pi}\frac{\omega}{\omega^2 - g^2/\pi} , \tag{2.26}$$

In the IR limit, $g^2 \to \infty$, the conductivity vanishes. This is consistent with the fact that the charge conservation symmetry has been gauged: in the absence of any kinetic term for the gauge field, it becomes a Lagrange multiplier fixing $j^\mu = 0$.

In the strict IR limit, it is of more use to consider the so-called irreducible conductivity, which is the response to the sum of internal and probe fields. We encountered this already as the self energy tensor, Eq. (2.13). Hence,

$$\sigma_{xx}^{1PI}(\omega) = \frac{i}{\omega}\Pi_{xx}(\omega, q = 0) = \frac{1}{\pi}\frac{i}{\omega} . \tag{2.27}$$

The ground state of the Schwinger model therefore has vanishing response to external probe fields, but the 1PI conductivity consists solely of a Drude peak.

---

[4]Note that the $U(1)_V$ of the free Dirac theory is gauged on coupling to $a_\mu$, meaning that the Schwinger model does not possess a continuous global symmetry. Therefore, $j^\mu$ is not a globally conserved current. Nevertheless, it is of interest to consider the response of $j^\mu$ to external probes, even though it does not correspond to a global charge.

# 3 Anomaly constraints on Fermi surfaces coupled to critical bosons

## 3.1 The microscopic model

We now proceed to discuss the role of anomalies in constraining a class of theories that have been widely used to describe metallic quantum critical points. Consider a Fermi surface of non-relativistic fermions, $\psi$, coupled to a fluctuating bosonic order parameter, $\phi_a$, $a = 1, \ldots, N_b$, in $d = 2$ spatial dimensions,

$$S = S_\psi + S_{\text{int}} + S_\phi \tag{3.1}$$

$$S_\psi = \int_{t,\boldsymbol{k}} \psi^\dagger (i\partial_t - \epsilon(\boldsymbol{k}))\psi \,, \tag{3.2}$$

$$S_{\text{int}} = \int_{\Omega,\boldsymbol{q}} \int_{\omega,\boldsymbol{k}} g^a(\boldsymbol{k}) \, \phi_a(\boldsymbol{q}, \Omega) \, \psi^\dagger(\boldsymbol{k} + \boldsymbol{q}, \omega + \Omega) \, \psi(\boldsymbol{k}, \omega) \,, \tag{3.3}$$

$$S_\phi = \frac{1}{2} \int_{t,\boldsymbol{x}} \left[ \lambda \, (\partial_t \phi_a)(\partial_t \phi^a) - m_c^2 \, \phi_a \phi^a - J \, (\boldsymbol{\nabla}\phi_a) \cdot (\boldsymbol{\nabla}\phi^a) + \cdots \right] \tag{3.4}$$

where $\int_{\boldsymbol{k},\omega} \equiv \int \frac{d^2\boldsymbol{k}\,d\omega}{(2\pi)^3}$, $\int_{t,\boldsymbol{x}} \equiv \int dt\,d^2\boldsymbol{x}$, and we continue to adopt a convention where repeated indices are summed. Here $g^a(\boldsymbol{k})$ is a coupling which varies along the Fermi surface, $\lambda, J$ are coupling constants, we choose the mass $m_c^2$ to tune the boson to criticality, and the $\cdots$ contains higher derivatives of $\phi$. Note that repeated indices will be summed over. Since we are working with non-relativistic fermions, we will use the standard Euclidean metric for raising and lowering the spatial indices and boson flavor indices. We will also adopt a Fourier transform convention where $f(t, \boldsymbol{x}) = \int \frac{d\omega\,d^2\boldsymbol{k}}{(2\pi)^3} e^{-i\omega t + i\boldsymbol{k}\cdot\boldsymbol{x}} f(\omega, \boldsymbol{k})$.

Our motivation for introducing multiple boson fields, $N_b \geq 1$, with flavor-dependent coupling $g^a$, is to accomodate a broad class of models of physical interest. For example, taking $N_b = 1$ and $g(\boldsymbol{k}) = g_0(\cos k_x - \cos k_y)$ results in a theory in the Ising-nematic universality class. On the other hand, taking $N_b = 2$, $a = 1, 2$ to be a spatial index, and $\epsilon(\mathbf{k}) = k^2/(2m_*)$, $g^a(\boldsymbol{k}) = k^a/m_*$ results in a theory of a fluctuating gauge field coupled to a circular Fermi surface, such as a spinon Fermi surface or the Halperin-Lee-Read (HLR) theory of the half-filled Landau level[5]. Also of interest to us is the case of "loop-current" order in a system with inversion symmetry, where the order parameter, and therefore $g^a(\boldsymbol{k})$, is odd under inversion. If there is additionally a $C_4$ rotation symmetry, then the order parameter must be a $N_b = 2$

---

[5]We can choose to work in temporal gauge where there is no temporal component of the gauge field. Note that for a gauge theory in temporal gauge, the kinetic term for the boson should be replaced with the proper gauge fixed Maxwell and, in the case of the HLR theory, Chern-Simons terms. This difference will not affect our general results.

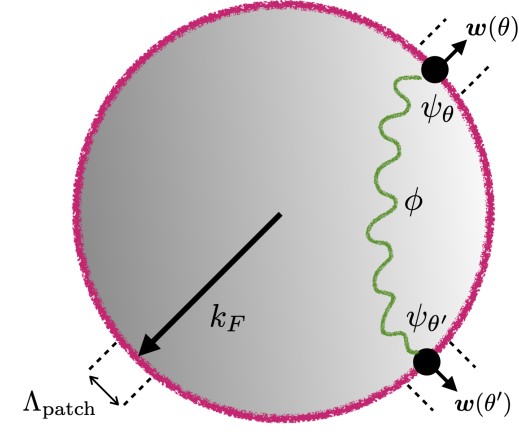

**Figure 1:** Schematic picture of the patch decomposition and "mid-IR" effective theory. We divide the Fermi surface into patches indexed by angles $\theta = 2\pi/N_{\text{patch}}, \ldots 2\pi$, each of width $\Lambda_{\text{patch}} \ll k_F$. Each patch is associated with a fermion field, $\psi_\theta$, and a unit normal, $\boldsymbol{w}(\theta)$. We couple the Fermi surface to a bosonic field, $\phi_a$, which can scatter fermions within a particular patch. While we neglect scattering of a fermion from one patch into another, we still allow the boson to mediate interactions *between* all of the patches (denoted by the wavy line).

component vector [62]. This is similar to the case of a fluctuating gauge field except that $g^a(\mathbf{k})$ can take a more general form, subject only to the constraint of $C_4$ symmetry. Finally, one could also consider the case of $N_f$ fermion flavors, but this will not affect our general results provided that the boson-fermion coupling does not dependent on the fermion flavor index. We discuss large-$N_f$ theories in more detail in Sections 5 and 6.

## 3.2 Patch decomposition and "mid-IR" effective theory

At criticality, models of the type in Eq. (3.1) have dynamics that are notoriously challenging to control, but it is possible to draw some basic conclusions about their low energy properties. The dangerous fluctuations of the bosons will be very long wavelength modes with $|\boldsymbol{q}| \ll k_F$, and we will focus on these exclusively. This suggests that it is legitimate to divide the Fermi surface into a large number, $N_{\text{patch}}$, of "patches" of width $\Lambda_{\text{patch}} \ll k_F$ (see Figure 1), such that scattering of a fermion from one patch into another can be regarded as unimportant compared with intra-patch scattering and can be discarded [17, 24, 29, 30].

While such a patch decomposition is a traditional way to view the problem of a Fermi surface coupled to critical bosons, one normally would make the stronger assumption that each pair of antipodal patches is *completely decoupled* from all other pairs [17, 24, 26, 29, 30].

The justification for this assumption would be that with interpatch scattering discarded, the only way for two patches to interact is through the boson. But the boson at wavevector $\boldsymbol{q}$ is believed to couple most strongly to those patches on the Fermi surface to which $\boldsymbol{q}$ is tangent. However, in this work, we will be interested in properties in the limit of vanishing boson momentum, such as the optical conductivity. At $\boldsymbol{q} = 0$, the boson can mediate interactions between *all* of the patches of the Fermi surface, so such a strict decoupling assumption would not make much sense. Thus, we will only assume that there are no inter-patch scattering terms, while allowing for the possibility that different patches could still interact through the boson.

Therefore, we seek to describe the low-energy properties of the "microscopic" theory in Eq. (3.1) through the action,

$$S = S_{\text{boson}} + \sum_\theta S_{\text{patch}}(\theta), \tag{3.5}$$

where the sum is over discrete patches labelled by $\theta = 2\pi/N_{\text{patch}}, \ldots, 2\pi$. Here $S_{\text{boson}}$ contains all the terms in Eq. (3.4), although we emphasize that we have only retained terms that are quadratic in the boson field and dropped any self-interactions. The action for the fermions in a single patch can be written in position space as

$$S_{\text{patch}}(\theta) = \int_{t,\boldsymbol{x}} \left[ \psi_\theta^\dagger \Big\{ i\partial_t + iv_F(\theta)[\boldsymbol{w}(\theta) \cdot \boldsymbol{\nabla}] + \kappa_{ij}(\theta)\partial_i\partial_j \Big\} \psi_\theta + g^a(\theta)\phi_a\, \psi_\theta^\dagger\psi_\theta \right], \tag{3.6}$$

where $v_F(\theta)$ is the Fermi velocity at patch $\theta$; $g^a(\theta)$ is the coupling $g^a(\boldsymbol{k})$ lying in the patch, which we take to be constant in each patch; $\boldsymbol{w}$ is the outward-pointing unit vector normal to the Fermi surface; and $\kappa_{ij}(\theta) = \partial_{\boldsymbol{k_i}}\partial_{\boldsymbol{k_j}}\epsilon|_{\boldsymbol{k}_F(\theta)}$ is the curvature tensor, which we take to only include the curvature of the Fermi surface. Note that we drop the curvature of the fermion dispersion in the direction perpendicular to the Fermi surface, i.e. we take $w_i(\theta)\kappa_{ij}(\theta) = 0$, since it is irrelevant (in the renormalization group sense) compared to the term linear in $w_i(\theta)\partial_i$, which has fewer derivatives. Note that $v_F(\theta)$, $g^a(\theta)$, $w_i(\theta)$ and $\kappa_{ij}(\theta)$ can all vary between different patches.

We emphasize that each patch comes with its own set of fermion fields, $\psi_\theta$, but they all share the same boson fields, $\phi_a$. In other words, we write the generating path integral defining the theory as

$$Z = \int \mathcal{D}\phi_a\, e^{iS_{\text{boson}}} \prod_\theta Z_{\text{patch}}[\phi_a; \theta]\,, \quad Z_{\text{patch}}[\phi_a; \theta] = \int \mathcal{D}\psi_\theta^\dagger \mathcal{D}\psi_\theta\, e^{iS_{\text{patch}}(\theta)}\,. \tag{3.7}$$

To avoid introducing additional degrees of freedom compared with the original microscopic action, it is necessary to impose a momentum cutoff, $\Lambda_{\text{patch}}$, for the fermions in each patch

in the direction parallel to the Fermi surface, such that the sum of the cutoffs over all the patches is equal to the length in momentum space of the original Fermi surface.

Since the action in Eqs. (3.5) – (3.6) retains all the terms in the original microscopic action that are believed to be important at low energies, the expectation is that it flows towards the same IR fixed point under the renormalization group as the microscopic action. We will refer to the action Eqs. (3.5) – (3.6) as specifying the "mid-IR theory." The majority of the conclusions of the present paper will be exact with respect to this mid-IR theory. It is therefore reasonable to expect that they will also be valid for the universal IR physics of the microscopic model, Eq. (3.1).

### 3.3   Symmetries and anomalies of the mid-IR theory

The mid-IR effective theory in Eq. (3.5) appears to lead to conservation of charge in each patch separately, as there are no inter-patch scattering terms, suggesting that the Lagrangian is invariant under independent U(1) rotations of fermions in each patch. At finite $N_{\text{patch}}$, we will therefore use the notation " U(1)$_{\text{patch}}$ " for this symmetry. However, we remark that giving a precise defininition of the global symmetry group corresponding to this conservation law is a somewhat subtle issue due to issues associated with slicing the Fermi surface up into patches. What we can say precisely is that in the $N_{\text{patch}} \to \infty$ limit, the global symmetry becomes the loop group, LU(1) (for a more detailed discussion, see Ref. [41]).

Like the conservation of left and right-moving fermions discussed in Section 2, the conservation of patch charge does not survive quantization: it is anomalous. This may be understood intuitively because the fermion in a single patch is essentially a 1D chiral fermion, in that its dispersion is linear in the momentum perpendicular to the Fermi surface. Therefore, each patch inherits the 1D chiral anomaly, such that applying an electric field would violate charge conservation in each individual patch. In fact, the patch charge is generally non-conserved even without an external field. The reason is that the boson field, $\phi$, is already playing a role analogous to a dynamical gauge field. To see this, note that coupling a dynamical vector potential, $\boldsymbol{a}$, to the action of a single patch, Eq. (3.6), would entail coupling to the current perpendicular to the Fermi surface by adding the term

$$S_{\text{gauge}}(\theta) = \int_{\boldsymbol{x},\tau} \boldsymbol{a} \cdot v_F(\theta) \boldsymbol{w}(\theta) \, \psi_\theta^\dagger \psi_\theta \,, \tag{3.8}$$

which exactly corresponds to the boson-fermion coupling term in Eq. (3.6) if we identify

$$\boldsymbol{a} \cdot \boldsymbol{w}(\theta) = \frac{1}{v_F(\theta)} \, g^a(\theta)\phi_a \,. \tag{3.9}$$

Note that, since $v_F(\theta)$, $g^a(\theta)$ and $\boldsymbol{w}(\theta)$ can be patch dependent, if we think of all of the patches together simultaneously, then the boson *does not* necessarily couple as a U(1) gauge field. It can still, however, be thought of as (a particular configuration of) a gauge field of the larger U(1)$_{\text{patch}}$ symmetry.

To motivate the precise anomaly equation, we first switch off the boson-fermion interactions and couple the theory to a background gauge field, $\mathbf{A}$. Since we now have a purely free fermion action, we can think the anomaly just in terms of the semi-classical equations of motion for single electrons,

$$\frac{d\boldsymbol{k}}{dt} = \boldsymbol{E} = -\partial_t \boldsymbol{A} \,. \tag{3.10}$$

Due to the chirality of the dispersion, the semiclassical flow of electrons in response to an electric field leads to a net inflow of electrons onto the Fermi surface. From this picture, we can calculate the anomaly equation for a patch,

$$\partial_t n_\theta + \boldsymbol{\nabla} \cdot \boldsymbol{j}_\theta = -\frac{\Lambda_{\text{patch}}}{(2\pi)^2} \, \boldsymbol{w}(\theta) \cdot \partial_t \boldsymbol{A} \,, \tag{3.11}$$

where $n_\theta$ and $\boldsymbol{j}_\theta$ are the charge and current densities of the patch, $\theta$, and we recall that $\Lambda_{\text{patch}}$ was the UV cutoff that we introduced for the patch in the direction parallel to the Fermi surface, i.e. it is the "width" of the patch in momentum space. In the limit as we take the number of patches $N_{\text{patch}} \to \infty$, the anomaly equation agrees with the LU(1) anomaly discussed in Ref. [41]. Notice that at $\boldsymbol{q} = 0$, using the fact that the current perpendicular to the Fermi surface is $v_F(\theta) n_\theta$, the anomaly directly implies the Drude conductivity for a free Fermi gas on summing over patches (analogous to the free Dirac example in 1D).

Now we set the background field to zero, but switch on the boson-fermion interaction terms. Since the boson couples (within an individual patch) like a dynamical gauge field, the anomaly equation in a patch is just determined by replacing $\boldsymbol{A} \to \boldsymbol{a}$, with $\boldsymbol{a}$ satisfying Eq. (3.9). Thus, we find

$$\partial_t n_\theta + \boldsymbol{\nabla} \cdot \boldsymbol{j}_\theta = -\frac{\Lambda_{\text{patch}}}{(2\pi)^2} \frac{1}{v_F(\theta)} \, g^a(\theta) \, \partial_t \phi_a \,. \tag{3.12}$$

We emphasize that, as in the Schwinger model discussed in Section 2.2, the anomaly equation holds in the quantum theory as an operator equality. Note that the anomaly can be viewed as arising from the transformation properties of the Jacobian of the fermion path integral, $\prod_\theta Z_{\text{patch}}[\phi_a; \theta]$, under a generalization of the axial rotations discussed in Section 2. From this analysis one finds that each patch contributes to the total anomaly as in Eq. (3.12). See Appendix B for an explicit derivation.

From Eq. (3.12), it can be immediately seen that there is still a conserved quantity associated with each patch, namely

$$\widetilde{n}_\theta = n_\theta + \frac{\Lambda_{\text{patch}}}{(2\pi)^2} \frac{1}{v_F(\theta)} \, g^a(\theta) \, \phi_a \,, \tag{3.13}$$

This is consistent with the result from Ref. [41] that any compressible metal should have an infinite-dimensional emergent symmetry group. Hence, the emergent symmetry group in the low-energy limit, in which we can take $N_{\text{patch}} \to \infty$, is still LU(1), but with a deformed set of generators.

Here we pause to make a technical point regarding regularization of the mid-IR effective theory, Eq. (3.6). Thus far, we have only specified a hard cutoff for the patch size, $\Lambda_{\text{patch}}$. A more precise definition of the UV regularization is necessary in order to determine the relation between the gauge invariant charge density operator, $n_\theta$, appearing in Eq. (3.12), and the operator, $\psi_\theta^\dagger \psi_\theta$, in the mid-IR theory. Below, we choose a regularization in which there is a cutoff for the momenta perpendicular to the Fermi surface, $\Lambda_\perp \ll k_F$, and a cutoff for frequency, $\Lambda_\omega \ll k_F$, with the requirement that $\Lambda_\perp \to \infty$ before $\Lambda_\omega \to \infty$. In this regularization, we may identify $\psi_\theta^\dagger \psi_\theta$ with $n_\theta$ (there is a subtlety when density probes are introduced, which we will comment on in Section 3.6). This regularization choice is common in perturbative approaches to the problem (see e.g. Ref. [30]) and corresponds to taking momentum integrals prior to frequency integrals when evaluating Feynman diagrams. In the opposite order of limits, where $\Lambda_\omega \to \infty$ first, it is in fact the conserved density, $\widetilde{n}_\theta$, that is identified with $\psi_\theta^\dagger \psi_\theta$. In Appendix A, we describe in more detail how this alternateive regularization choice modifies some formal statements but ultimately leads to the same physics.

## 3.4 The boson propagator at $\mathbf{q} = 0$

Unlike the Schwinger model in Section 2.2, the anomaly equation in Eq. (3.12) depends on components of the current operator $J^i(\omega, \boldsymbol{q})$ parallel to the Fermi surface. As a result, $n_\theta(\omega, \boldsymbol{q})$ cannot be expressed as a function of $\phi_a(\omega, \boldsymbol{q})$ alone and the anomaly does not fix the full boson propagator as a function of $\omega$ and $\boldsymbol{q}$. Nevertheless, the anomaly determines the boson propagator at zero wave vector, $\boldsymbol{q} = 0$. The argument proceeds very similarly to the case of the Schwinger model. At $\boldsymbol{q} = 0$, the anomaly equation Eq. (3.12) becomes

$$\frac{dn_\theta}{dt} = -\frac{\Lambda_{\text{patch}}}{(2\pi)^2} \frac{g^a(\theta)}{v_F(\theta)} \frac{d\phi_a}{dt} \,, \tag{3.14}$$

where all quantities are now evaluated at $\boldsymbol{q} = 0$, and we remind the reader that we implicitly sum over repeated indices (except $\theta$). Meanwhile, in the presence of a spatially uniform source term for the boson field,

$$S_{\text{source}} = \int_{\boldsymbol{x},t} h^a(t)\, \phi_a(t, \boldsymbol{x})\,, \tag{3.15}$$

the equation of motion for the boson is

$$\left(\lambda \frac{d^2}{dt^2} + m_c^2\right)\phi^a = h^a + \sum_\theta g^a(\theta)\, n_\theta\,, \tag{3.16}$$

where we have recalled the action given in Eqs. (3.4) – (3.6). By similar arguments to Section 2.2, this actually holds as an operator equality in the quantum theory. Taking the time derivative of Eq. (3.16), passing to frequency space, and invoking Eq. (3.14), we find

$$\left[\left(-\lambda\omega^2 + m_c^2\right)\delta^{ab} + \Pi^{ab}\right]\phi_b(\omega) = h^a(\omega), \tag{3.17}$$

where we defined

$$\Pi^{ab} = \sum_\theta \frac{\Lambda_{\text{patch}}}{(2\pi)^2}\frac{g^a(\theta)g^b(\theta)}{v_F(\theta)}\,. \tag{3.18}$$

By taking expectation values of Eq. (3.17), we can read off the boson propagator at $\boldsymbol{q} = 0$, via linear response theory, $\langle\phi(\omega)\rangle = D_{ab}(\omega)\, h^b(\omega)$. We obtain[6]

$$D_{ab}(\omega, \boldsymbol{q} = 0) = i\langle\phi_a(-\omega, \boldsymbol{q} = 0)\phi_b(\omega, \boldsymbol{q} = 0)\rangle = \left[\left(-\lambda\omega^2 + m_c^2\right)\mathbb{I} + \Pi\right]^{-1}_{ab}. \tag{3.19}$$

Since $-iD_0(\omega) = i[\lambda\omega^2 - m_c^2]^{-1}$ is just the bare boson propagator at $\boldsymbol{q} = 0$, we see that $\Pi^{ab}$, as defined in Eq. (3.18), is precisely the boson self energy, $\Pi^{ab}(\omega, \boldsymbol{q} = 0)$ (see Figure 2). Crucially, the boson self energy is *independent* of frequency. This result, which is a direct consequence of the anomaly, Eq. (3.12), is a completely non-perturbative statement about the mid-IR theory. Among other things, it will ultimately enable us to completely fix the optical conductivity. Thus, Eqs. (3.18) – (3.19) are among the main results of this work.

The simple, frequency-independent result for the boson self energy in Eq. (3.18) is surprising given the strong coupling of the Fermi surface to critical fluctuations. Indeed, the self energy we obtain is precisely the value of the one-loop contribution alone! In other words, this means that, at least for $\Pi^{ab}(\omega, \boldsymbol{q} = 0)$, the random phase approximation (RPA) limit produces the exact result. This contrasts with earlier results using various methods

---

[6]Throughout the manuscript we use the notation, e.g. $\langle\phi_a(-\omega, \boldsymbol{q} = 0)\phi_b(\omega, \boldsymbol{q} = 0)\rangle$, for real frequency correlation functions, although these more precisely denote the Fourier transform of $\langle[\phi_a(t), \phi_b(0)]\rangle\,\Theta(t)$.

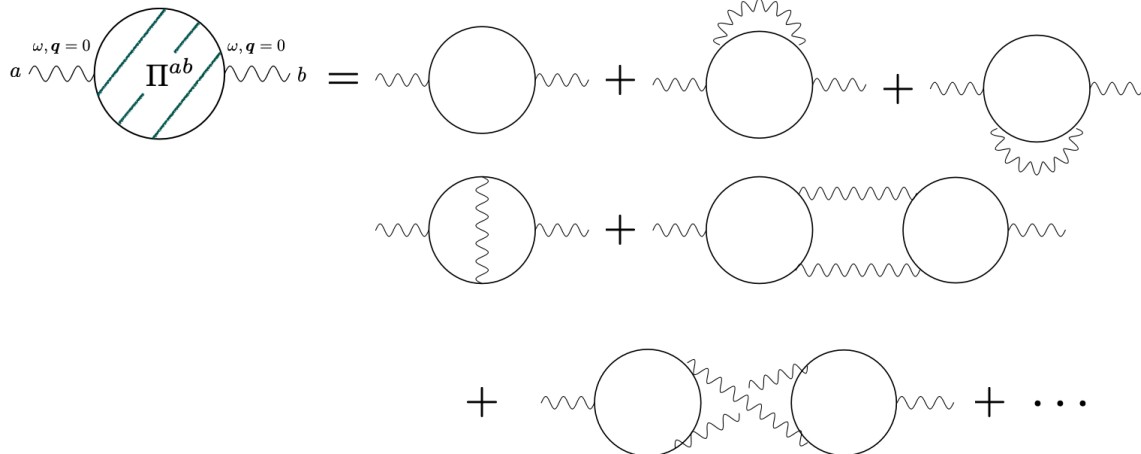

**Figure 2:** A sample of the Feynman diagrams that contribute to the boson self energy, $\Pi^{ab}(\omega, \boldsymbol{q} = 0)$, where solid and wavy lines denote bare fermion and boson propagators, respectively. The boson self energy is naïvely constrained by (intra-patch) charge conservation alone, but the presence of the anomaly deforms this constraint. The resulting Ward identity fixes this infinite series of diagrams non-perturbatively. This leads to the result in Eq. (3.4), which is equal to the one-loop contribution to the sum.

[27, 37, 54, 55], which have suggested frequency scaling of $\Pi^{ab}(\omega, \boldsymbol{q} = 0)$ at the fixed point (i.e. with irrelevant operators set to zero). We will consider these earlier approaches in more detail in Section 5. However, from the point of view of anomalies, our result is quite natural: it is simply a consequence of the statement that the $\mathrm{U}(1)_{\mathrm{patch}}$ anomaly, like the chiral anomaly in the Schwinger model discussed in Section 2.2 (as well as analogous anomalies in higher dimensions), is one-loop exact.

At the critical point (where $m_c^2 = 0$, as we demonstrate explicitly in Section 3.6), Eq. (3.19) shows that the boson at $\boldsymbol{q} = 0$ is gapped out, and it oscillates at frequencies $\omega_P$ given by the eigenvalues of $\lambda^{-1}\Pi$. These gapped modes closely resemble the plasmon in the Schwinger model encountered in Section 2.2, and for the case where $\phi$ is a U(1) gauge field would correspond to the usual electromagnetic plasmon[7]. However, care needs to be taken in interpreting this result more broadly. It has been derived from the mid-IR effective theory, which was justified by the expectation that it will flow to the correct IR fixed-point theory. But since the "plasmons" are at finite frequency, they are not an IR property of the

---

[7]For bosons with non-local kinetic terms, as in the example of HLR theory with $1/r$ Coulomb interactions, there is a gapless plasmon mode in the sense that the dispersion $\omega_{\boldsymbol{q}}$ approaches zero as $|\boldsymbol{q}| \to 0$. This does not conflict with the result from the anomaly, since the latter is obtained at $\boldsymbol{q}$ exactly equal to zero.

system.

Therefore, it is unclear to what extent they represent an actual oscillation mode of the original microscopic system. By contrast, the IR limit of Eq. (3.19) is obtained by neglecting the $\lambda\omega^2$ term in the denominator. Then we find that the boson propagator in the IR limit is finite and frequency-independent, which should be true also for the original microscopic theory, since it is an IR property.

## 3.5   The optical conductivity

Having fixed the boson self energy $\Pi^{ab}$, we are now prepared to show that the anomaly completely determines the optical conductivity,

$$\sigma^{ij}(\omega) = \frac{K^{ij}(\omega, \boldsymbol{q}=0)}{-i\omega} = \frac{i\langle J^i(-\omega, \boldsymbol{q}=0) J^j(\omega, \boldsymbol{q}=0)\rangle}{-i\omega} , \qquad (3.20)$$

in the IR limit where all irrelevant operators are dropped. We will see below that the inclusion of certain irrelevant scattering processes can introduce corrections not constrained by the anomaly, even though the boson propagator at $\boldsymbol{q}=0$ remains completely fixed.

We have defined $J^i = \sum_\theta j_\theta^i$ to be the physical EM current summed over patches. Inspecting Eqs. (3.5) and (3.6), then by minimal coupling to the spatial component of a gauge field, we see that this current can be decomposed into components that are perpendicular and parallel to the Fermi surface,

$$J^i = J^i_\perp + J^i_\parallel , \qquad (3.21)$$

$$J^i_\perp = \sum_\theta v_F(\theta)\, w^i(\theta)\, n_\theta = \sum_\theta v_F(\theta)\, w^i(\theta)\, \psi_\theta^\dagger \psi_\theta , \qquad (3.22)$$

$$J^i_\parallel = \sum_\theta \frac{1}{2} \kappa^{ij}_{\text{FS}}(\theta) \left[ i\psi_\theta^\dagger \partial_j \psi_\theta - i\partial_j \psi_\theta^\dagger \, \psi_\theta \right] , \qquad (3.23)$$

where we have defined the curvature of the Fermi surface as

$$\kappa_{\text{FS}}(\theta) = Y_\theta\, \kappa(\theta)\, Y_\theta , \;\; Y^{ij}_\theta = \delta^{ij} - w^i(\theta)\, w^j(\theta) , \qquad (3.24)$$

with $\kappa^{ij}(\theta)$ having been defined in Eq. (3.6). On the other hand, the curvature of the *dispersion* leads to irrelevant corrections to $J_\perp$ which we set to zero at the outset.

We argue that, in the strict low energy limit, $J_\perp$ contributes to the conductivity, but $J_\parallel$ does not. This is a consequence of the fact that the contribution of each patch to $J_\parallel$ is proportional to the momentum parallel to the Fermi surface along that patch,

$$\mathcal{P}^i_\parallel(\theta) = \frac{1}{2} Y^{ij}_\theta \left[ i\psi_\theta^\dagger \partial_j \psi_\theta - i\partial_j \psi_\theta^\dagger \, \psi_\theta \right] . \qquad (3.25)$$

Were the momentum in each pair of antipodal patches, $\mathcal{P}(\theta) + \mathcal{P}(\theta + \pi)$, exactly conserved, then the contribution of $J_\parallel$ to the conductivity in Eq. (3.20) would vanish. Although this is not an exact property of the mid-IR effective theory we have written in Eq. (3.5), it should be viewed as an emergent property of the ultimate IR fixed point. At low energies, each patch only couples strongly to bosons with momentum, $\boldsymbol{q}$, tangent to the Fermi surface but weakly to all other boson momenta [17, 26, 29]. Indeed, all of the critical (i.e. gapless) boson fluctuations, which have $\omega \ll |\boldsymbol{q}|$, only couple strongly to a single antipodal pair of patches (although bosons at $\boldsymbol{q} = 0$ couple equally to each patch, a fact which will become important later in this Section). Furthermore, in the IR the boson does not contribute to the total momentum operator: the frequency-dependent term in the boson Lagrangian, $\lambda(\partial_t \phi)^2$, vanishes in the IR limit because it is irrelevant compared to the frequency-dependent terms generated by interactions with the fermions. Therefore, the dynamics of the mid-IR theory respect an emergent momentum conservation for each pair of antipodal patches, and we are led to conclude that $J_\parallel$ cannot contribute to the conductivity in the IR limit but *can* contribute to the conductivity at scales where such inter-patch interactions appear.

If $J_\parallel$ has vanishing contribution to the conductivity, to compute the current-current correlator, $K_{ij}(\omega)$, we simply introduce a probe electric field via a spatially uniform vector potential coupling to $J_\perp$,

$$S_{\text{probe}} = \int_{\boldsymbol{x},t} A_i(t)\, J_\perp^i(t, \boldsymbol{x})\,. \tag{3.26}$$

The anomaly equation (3.14) is now modified by the presence of the probe to

$$\frac{dn_\theta}{dt} = -\frac{\Lambda_{\text{patch}}}{(2\pi)^2}\left[\frac{g^a(\theta)}{v_F(\theta)}\frac{d\phi^a}{dt} + w^i(\theta)\frac{dA_i}{dt}\right]\,. \tag{3.27}$$

The anomaly equation, Eq. (3.27), allows us to establish an operator relation[8],

$$j^i(\omega) = -\sum_\theta \frac{\Lambda_{\text{patch}}}{(2\pi)^2}\, w^i(\theta)\left[g^a(\theta)\phi_a(\omega) + v_F(\theta)w^j(\theta)A_j(\omega)\right]\,. \tag{3.28}$$

Here and throughout this subsection, all fields are evaluated at $\boldsymbol{q} = 0$. This means we can represent correlation functions of $j^i(\omega)$ in terms of correlation functions of $g^a(\theta)\phi_a(\omega)$ by substituting Eq. (3.28) into Eq. (3.26). However, this substitution cannot be performed

---

[8]One might worry that this equation is in conflict with gauge invariance: the left hand side involves the EM current, which is gauge invariant, while the right hand side involves the gauge fields themselves. However, in coupling only to vector potentials, we have implicitly fixed to temporal gauge, $A_t = 0$, so Eq. (3.27) should be interpreted in the context of this particular gauge.

directly: the presence of the background field in Eq. (3.28) indicates that the operator identity as $A_j \to 0$ only holds up to contact terms. We can see this by imposing the requirement that at finite field we recover the anomaly equation as the one-point function,

$$\langle j^i(\omega) \rangle_A = -i \frac{\delta}{\delta A_i(-\omega)} \log Z[A]$$

$$= -\sum_\theta \frac{\Lambda_{\text{patch}}}{(2\pi)^2} w^i(\theta) \left[ g^a(\theta) \langle \phi_a(\omega) \rangle_A + v_F(\theta) w^j(\theta) A_j(\omega) \right]. \qquad (3.29)$$

To satisfy this requirement, we must add a "diamagnetic" contact term for the probe,

$$S_{\text{dia}} = -\frac{1}{2} \sum_\theta \frac{\Lambda_{\text{patch}}}{(2\pi)^2} v_F(\theta) w^i(\theta) w^j(\theta) \int_\omega A_i(-\omega) A_j(\omega), \qquad (3.30)$$

i.e. we perform the substitution,

$$S_{\text{probe}} = \int_\omega A_i(-\omega) j^i(\omega) \to -\int_\omega A_i(-\omega) \left( \sum_\theta \frac{\Lambda_{\text{patch}}}{(2\pi)^2} w^i(\theta) \, g^a(\theta) \phi_a(\omega) \right) + S_{\text{dia}}. \qquad (3.31)$$

Thus, the anomaly allows us to immediately compute the optical conductivity in terms of the boson propagator at $\boldsymbol{q} = 0$,

$$\sigma^{ij}(\omega) = \frac{1}{-i\omega} i \frac{\delta}{\delta A_i(-\omega)} \frac{\delta}{\delta A_j(\omega)} \log Z[A] \Big|_{A=0}$$

$$= \frac{i}{\omega} \left[ \frac{\mathcal{D}^{ij}_{(0)}}{\pi} - V^{a,i} V^{b,j} \, i \langle \phi_a(-\omega) \phi_b(\omega) \rangle \right], \qquad (3.32)$$

where we have defined

$$\mathcal{D}^{ij}_{(0)} = \sum_\theta \frac{\Lambda_{\text{patch}}}{4\pi} v_F(\theta) \, w^i(\theta) \, w^j(\theta), \qquad (3.33)$$

which is the Drude weight of the non-interacting Fermi surface written as a sum over patches, as well as the vertex factors,

$$V^{a,i} = \sum_\theta \frac{\Lambda_{\text{patch}}}{(2\pi)^2} g^a(\theta) \, w^i(\theta) \qquad (3.34)$$

As we show explicitly in Section 3.6, criticality occurs as $m_c^2 \to 0$ and $\chi_{\phi_a \phi_a} = \frac{1}{m_c^2} \to \infty$. Substituting Eq. (3.19) and $m_c^2 = 0$, we find the optical conductivity at criticality to be,

$$\sigma^{ij}(\omega) = \frac{i}{\omega} \left[ \frac{\mathcal{D}^{ij}_{(0)}}{\pi} - V^{a,i} V^{b,j} \left( \frac{1}{-\lambda\omega^2 + \Pi} \right)_{ab} \right], \qquad (3.35)$$

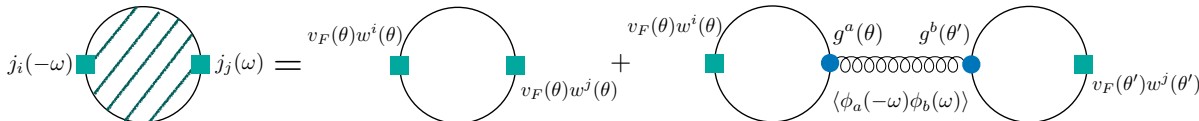

**Figure 3:** Diagrammatic expression for the fixed point current correlator, $\langle j_i(-\omega, \boldsymbol{q} = 0) j_j(\omega, \boldsymbol{q} = 0) \rangle$, obtained in Eqs. (3.35) and (3.36). Here we define boxes as current insertions, which introduce factors of $v_F(\theta) w^i(\theta)$, and circles as boson-fermion vertices, which introduce factors of $g^a(\theta)$. Solid lines denote bare fermion propagators, and the curly line denotes the full boson propagator including all quantum corrections, which we determined using the $U(1)_{\text{patch}}$ anomaly in Eq. (3.19). Because at zero momentum the boson couples strongly to each of the patches, every closed fermion loop involves a sum over the patch index, $\theta$. Computing these diagrams reproduces the expression we obtained in Eq. (3.35), meaning that all additional corrections involving internal boson propagators cancel.

with $\Pi$ given by Eq. (3.18). Note that here we have used the fact that (in our chosen regularization) the critical point corresponds to $m_c = 0$, as we will show in Section 3.6. Eq. (3.35) is another central result of this work, constituting a *non-perturbative* calculation of the conductivity of the IR fixed point theory, with irrelevant scattering processes set to zero (See Figure 3 for an interpretation of Eq. (3.35) in terms of Feynman diagrams).

Observe from Eq. (3.35) that the conductivity remains non-dissipative. Indeed, the real part of the conductivity is just a sum of delta functions coming from the poles occuring at $\omega = 0$ and whenever $\omega^2$ is equal to an eigenvalue of $\lambda^{-1}\Pi$. The latter poles are related to the plasmon oscillations discussed in Section 3.4, but since these live high in the spectrum of the mid-IR theory and do not reflect IR physics, we will not consider them any further.

Taking the IR limit, in which $\lambda\omega^2$ is dropped compared to $\Pi$, setting $v_F(\theta) = v_F \equiv$ constant for simplicity, and defining $\text{Tr}_\theta[fg] \equiv \sum_\theta f(\theta)g(\theta)$, we find that at the IR fixed point the optical conductivity is given by a Drude form alone,

$$\sigma^{ij}(\omega) = \frac{i}{\omega} \frac{\Lambda_{\text{patch}}}{(2\pi)^2} v_F \left( \text{Tr}_\theta[w^i w^j] - \text{Tr}_\theta[g^a w^i] \left( \text{Tr}_\theta[gg] \right)^{-1}_{ab} \text{Tr}_\theta[g^b w^j] \right), \tag{3.36}$$

Remarkably, there is no frequency-dependent contribution related to quantum critical fluctuations. We therefore conclude that any low energy ($\lambda\omega^2 \to 0$) frequency dependence in

the conductivity must come from irrelevant operators that break the $U(1)_{\text{patch}}$ symmetry of the mid-IR theory explicitly, thus invalidating the anomaly equation. Nevertheless, unless the second term in Eqs (3.35) – (3.36) is vanishing due to symmetry, the coupling to the boson still corrects the Drude weight from that of a non-interacting Fermi surface, despite the fact that there is only forward (intra-patch) scattering.

The formulae in Eqs. (3.35) – (3.36) apply for general order parameters, $\phi^a$, and we comment here on particular examples of special interest. For systems possessing a $C_2$ inversion symmetry under which the boson field is inversion-even [meaning $g^a(\theta) = g^a(\theta + \pi)$], it follows that the contribution of each patch ($\theta$) to Eq. (3.34) cancels with its antipode ($\theta + \pi$), and $V^{a,i} = 0$. Therefore,

$$\sigma_{\text{Ising}}^{ij}(\omega) = \frac{\mathcal{D}_{(0)}^{ij}}{\pi} \frac{i}{\omega} . \tag{3.37}$$

A major example in this class is a Fermi surface coupled to an Ising-nematic order parameter. In this case, the result is identical to that for a decoupled Fermi surface. We remark that this statement is valid in the mid-IR theory. In the microscopic theory, the short-wavelength fluctuations of the boson can of course have the effect of renormalizing the Fermi velocity.

Another important class of examples involve Fermi surfaces coupled to U(1) gauge fields, such as composite Fermi liquids or spinon Fermi surfaces. In this case, the conductivity *vanishes* because the couplings $g^a(\theta)$ are replaced with $v_F(\theta)w^i(\theta)$, meaning that the two terms in Eq. (3.36) cancel. This is a consequence of the fact that gauge fluctuations constrain $j^i(\omega) = 0$ as $\lambda \to 0$, as we saw in the simple example of the Schwinger model. However, we can instead consider the so-called "irreducible" conductivity, which gives the response to the sum of the fluctuating and background fields. From Eq. (3.27), we immediately see this is simply the one-loop Drude weight,

$$\sigma_{\text{gauge}}^{ij}(\omega) = 0 \, , \, \sigma_{\text{gauge, 1PI}}^{ij}(\omega) = \frac{\Pi^{ij}(\omega)}{-i\omega} = \frac{i}{\omega} \frac{1}{\pi} \sum_{\theta} \frac{\Lambda_{\text{patch}}}{4\pi} \frac{g^i(\theta)g^j(\theta)}{v_F(\theta)} . \tag{3.38}$$

Therefore, even though the EM conductivity vanishes, the irreducible conductivity is nonzero, equal to its one-loop (RPA) form, and fixed completely by the anomaly.

Finally, we can consider a more general class of order parameters that are odd under inversions, with $g^a(\theta) = -g^a(\theta + \pi)$, among which U(1) gauge theories are a special example. We broadly refer to these as "loop current" order parameters. For such theories, the Drude weight *does not* generally vanish, and is instead reduced compared to $\mathcal{D}_{(0)}$ according to Eq. (3.36). We discuss physical intepretations and implications of this result and how it is related to the "critical drag" concept of Refs. [48, 49] in a companion paper, Ref. [63].

### 3.6 Static susceptibilities

#### 3.6.1 Boson susceptibility and approach to criticality

Here we derive the boson susceptibility and show that criticality in the mid-IR theory occurs at $m_c^2 = 0$ in our chosen UV regularization. This may appear surprising at first since the structure of Eq. (3.19) suggests that in order to reach criticality, we should tune $m_c^2$ to cancel the self energy contribution, $\Pi$. However, this naïve thinking is incorrect. The correct diagnostic of criticality is the divergence of the boson susceptibility,

$$\chi_{\phi_a \phi_b} = \lim_{\boldsymbol{q} \to 0} \lim_{\omega \to 0} D_{ab}(\boldsymbol{q}, \omega) \tag{3.39}$$

which is not evaluated in the same $\boldsymbol{q}, \omega$-regime as Eq. (3.19). Indeed, in this subsection, we will give general arguments fixing the static boson susceptibility, and we will see that criticality is achieved at $m_c^2 = 0$. Note that this holds for the paticular regularization we chose above in Section 3.3, wherein $\Lambda_\perp \to \infty$ faster than $\Lambda_\omega \to \infty$ (see Appendix A). This regularization respects a kind of gauge invariance for $g^a(\theta)\phi_a$, which ensures that criticality occurs at $m_c^2 = 0$. However, we emphasize that upon tuning to criticality, the form of the boson propagator will be regularization independent.

Lacking any non-perturbative means to compute at general $\boldsymbol{q} \neq 0$, we will simply compute the susceptibilty exactly at $\boldsymbol{q} = 0$, namely,

$$\chi_{\phi_a \phi_b} = \lim_{h \to 0} \frac{\delta \langle \phi_a \rangle_h}{\delta h^b}, \tag{3.40}$$

where $\langle \cdot \rangle_h$ denotes the expectation value taken with respect to an action with a static and spatially uniform source term, $h^a \phi_a$, $h^a \equiv$ constant, added. In other words, we are assuming that $\lim_{\omega \to 0} D_{ab}(\boldsymbol{q}, \omega)$ is continuous in $\boldsymbol{q}$ as $\boldsymbol{q} \to 0$. This is reasonable because we expect that if $\chi_{\phi_a \phi_b}$ is finite, then for very long wavelength modulations of the source term, locally the system will just reach an equilibrium at the local value of the source.

We begin from Eq. (3.16). If we take the source field $h^a$ to be independent of time, then the expectation value of $\phi$ is independent of time in equilibrium. Using the equation of motion, we find that

$$m_c^2 \langle \phi^a \rangle_h = h^a + \sum_\theta g^a(\theta) \langle n_\theta \rangle_h. \tag{3.41}$$

Note that here we assume the absence of self interactions for the boson field (as specified in our definition of the mid-IR theory in Section 3.2). From this we conclude that

$$m_c^2 \chi_{\phi_a \phi_b} = \delta_{ab} + \sum_\theta g^a(\theta) \chi_{n_\theta \phi_b}. \tag{3.42}$$

To compute the susceptibility on the right hand side, we introduce source fields $h^a$ and $\mu_\theta$, and consider the partition function

$$Z[h, \mu] = \int \mathcal{D}\phi \left[ \prod_\theta \mathcal{D}\psi_\theta^\dagger \mathcal{D}\psi_\theta \right] \exp \left\{ iS + i \int_{t, \boldsymbol{x}} \left[ h^a \phi_a + \sum_\theta \mu_\theta \, \psi_\theta^\dagger \psi_\theta \right] \right\}, \tag{3.43}$$

where $S$ is the action in Eq. (3.5). Next, we make a change of integration variables in the path integral in which the momentum of each fermion field perpendicular to the Fermi surface is shifted by a constant amount,

$$\psi_\theta(\boldsymbol{x}, t) \mapsto \exp\left( i\mu_\theta \, v_F(\theta)^{-1} \boldsymbol{w}(\theta) \cdot \boldsymbol{x} \right) \psi_\theta(\boldsymbol{x}, t). \tag{3.44}$$

We observe that this completely eliminates the dependence on $\mu_\theta$. Importantly, despite the existence of an anomaly, the integration measure transforms trivially under such a transformation because it is time-independent. Had we chosen a *time-dependent* transformation[9], the Jacobian would transform non-trivially and generate the anomaly. See Appendix B for more details on the transformation properties of the path integral measure.

We therefore conclude that $\chi_{n_\theta \phi_b} = 0$; hence we find

$$\chi_{\phi_a \phi_b} = \frac{\delta_{ab}}{m_c^2}. \tag{3.45}$$

In particular, we see that the critical point in which the boson susceptibility diverges occurs when the bare mass, $m_c$, goes to zero. This is perhaps most intuitive in the case of a Fermi surface coupled to a dynamical U(1) gauge field, where $m_c = 0$ is simply the condition for the action to be gauge invariant. Note that (3.45) for the boson susceptibility was, in fact, already obtained in Ref. [17]. However, the arguments used there is somewhat different from the one presented here.

### 3.6.2   Patch density susceptibilities and Fermi liquid parameters off criticality

We now consider the intra-patch density susceptibility, $\chi_{n_\theta n_\theta}$. Because the change of variables in Eq. (3.44) eliminates any dependence on the density probe, $\mu_\theta$, one may worry that $\chi_{n_\theta n_\theta}$ and, therefore, the compressibility vanishes. This would of course be physically incorrect,

---

[9]This is because we work in a regularization where $g^a(\theta)\phi_a$ is treated like the spatial component of a U(1)$_{\text{patch}}$ gauge field, and we have chosen a regularization that preserves gauge invariance for this field. We emphasize that this does not mean that $g^a(\theta)\phi_a$ couples like a physical EM vector potential. For example, an Ising-nematic order parameter couples like an *axial* vector potential to antipodal patches. See Appendix B for a more thorough discussion.

as any theory of a stable Fermi surface should be compressible. However, the apparent vanishing of $\chi_{n_\theta n_\theta}$ is simply a consequence of our particular choice of UV regularization, which preserves gauge transformations with arbitrary wave vector but not frequency. This is why it was possible to shift all of the fermion momenta in Eq. (3.44). In contrast, because of the presence of a finite $\Lambda_\omega$ as the limit $\Lambda_\perp \to \infty$ is taken, this regularization *does not* treat scalar potentials (i.e. temporal components of gauge fields) such as $\mu_\theta$ in a gauge invariant way. Indeed, it is straightforward to see using this regularization that at one loop $\chi_{\psi_\theta^\dagger \psi_\theta, \psi_\theta^\dagger \psi_\theta} = 0$ [30] (see Appendix A for a discussion of the regularization that uses the opposite order of limits). All of this indicates a need to revisit the association of the physical density, $n_\theta$, with $\psi_\theta^\dagger \psi_\theta$.

To repair the mid-IR theory in our chosen regularization and obtain a finite $\chi_{n_\theta n_\theta}$, it is necessary to add a screening mass term directly to the action for the probe, $\mu_\theta$,

$$S_{\text{screening}} = - \int_{t,\boldsymbol{x}} \sum_\theta \frac{1}{2} M^2(\theta)\, \mu_\theta^2 \,, \quad M^2(\theta) = \frac{\Lambda_{\text{patch}}}{(2\pi)^2\, v_F(\theta)}. \tag{3.46}$$

The particular value of $M^2(\theta)$ is fixed to restore gauge invariance under the physical EM gauge symmetry, wherein $\mu_\theta$ couples as the temporal component of a gauge field. The need for such a mass term indicates that in the presence of the probe, $\mu_\theta$, we should re-define

$$n_\theta = \psi_\theta^\dagger \psi_\theta + M^2(\theta)\, \mu_\theta \,, \tag{3.47}$$

which in turn yields

$$\chi_{n_\theta n_{\theta'}} = M^2(\theta)\, \delta_{\theta\theta'} = \frac{\Lambda_{\text{patch}}}{(2\pi)^2\, v_F(\theta)}\, \delta_{\theta\theta'} \,. \tag{3.48}$$

This result is another non-perturbative feature of the mid-IR theory, and we will give a brief derivation of $M^2(\theta)$ from the anomaly equation later in this Section.

It is also of interest to consider the susceptibility of the conserved $U(1)_{\text{patch}}$ charge density,

$$\widetilde{n}_\theta = n_\theta + \frac{\Lambda_{\text{patch}}}{(2\pi)^2} \frac{1}{v_F(\theta)}\, g^a(\theta)\, \phi_a \,, \tag{3.49}$$

introduced in Eq. (3.13). Using Eq. (3.48) along with the earlier results, $\chi_{n_\theta \phi_a} = 0$, $\chi_{\phi_a \phi_b} = \delta_{ab}/m_c^2$, we find

$$\chi_{\widetilde{n}_\theta \widetilde{n}_{\theta'}} = \frac{\Lambda_{\text{patch}}(\theta)}{(2\pi)^2 v_F(\theta)}\, \delta_{\theta\theta'} + \frac{\Lambda_{\text{patch}}(\theta)\, \Lambda_{\text{patch}}(\theta')}{(2\pi)^4} \frac{g^a(\theta) g_a(\theta')}{v_F(\theta) v_F(\theta')} \frac{1}{m_c^2} \,, \tag{3.50}$$

where we have allowed $\Lambda_{\text{patch}}(\theta)$ to vary in each patch for clarity of presentation. Like our earlier results, the above expression for the charge susceptibility, $\chi_{\widetilde{n}_\theta \widetilde{n}_{\theta'}}$, is *non-perturbative*

in the mid-IR theory. Furthermore, it is valid for any positive[10] value of $m_c^2$. As $m_c^2 \to 0$ and criticality is approached, $\chi_{\widetilde{n}_\theta \widetilde{n}_{\theta'}} \to \infty$ alongside the boson susceptibility. When $g^a(\theta)$ has a definite angular momentum, the susceptibility of $\chi_{\mathcal{O}\mathcal{O}}$ of a generic operator $\mathcal{O} = \sum_\theta f(\theta)\widetilde{n}(\theta)$ diverges if and only if $f(\theta)$ and $g^a(\theta)$ have the same angular momentum. Note finally that in the $N_{\mathrm{patch}} \to \infty$ limit of a continuous Fermi surface, parameterized by a continuous variable $\theta$, we replace $\Lambda_{\mathrm{patch}}(\theta) \to |\partial_\theta \boldsymbol{k}_F(\theta)|$ and $\delta_{\theta\theta'} \to \delta(\theta - \theta')$ in Eqs. (3.48) and (3.50).

The result for the susceptibility in Eq. (3.50) has interesting implications for the system off critcality in the disordered phase, where $m_c^2 > 0$. Away from criticality, we should expect that the system will ultimately flow to a Fermi liquid fixed point, and one may consider how the parameters of this Fermi liquid (the Fermi velocity and Landau parameters) diverge as criticality is approached. Like any Fermi liquid, this fixed point will have an emergent conserved charge associated with each point on the Fermi surface in the absence of external fields. As $\widetilde{n}_\theta$ is already conserved even in the mid-IR theory, it is natural to identify it with the Fermi surface charge distribution of the mid-IR theory off of criticality. Thus, the susceptibility of the $\widetilde{n}_\theta$ can be computed in terms of the Fermi liquid parameters; namely, the quasiparticle's renormalized Fermi velocity, $v_F^*(\theta)$, and the Landau interaction function, $F(\theta, \theta')$. Our general results do not allow us to determine the singularity of the renormalized Fermi velocity, and hence of the Landau interaction. Nevertheless, as we have just shown, the susceptibility of $\widetilde{n}_\theta$ is directly fixed exactly by general arguments in the mid-IR theory. Therefore, Eq. (3.50) constitutes a non-trivial constraint relating the renormalized Fermi velocity $v_F^*(\theta)$ and the Landau interaction function $F(\theta, \theta')$ as criticality is approached from the disordered phase.

The result for the singularity of the Fermi liquid susceptibility in Eq. (3.50) was in fact anticipated earlier by the perturbative calculations of Maslov and Chubukov [40], as well as the arguments of Mross $et$ $al.$ [30]. Our presentation considerably clarifies the theoretical underpinnings of these singular susceptibilities by relating them to exact non-perturbative properties of the mid-IR patch theory.

Finally, we remind the reader that our calculation of the Drude weight in Section 3.5 using the anomaly can be readily extended off criticality to the proximate Fermi liquid. Alternatively, we can calculate the Drude weight in the proximate Fermi liquid by relating it to various thermodynamic susceptibilities. Gratifyingly, it can be verified (see our upcoming

---

[10]In the ordered phase, the mid-IR theory predicts a negative susceptibility for the boson field, as can be seen from Eq. (3.45). This is because the mid-IR theory used here adequately describes the approach to criticality only from the disordered phase. To describe the ordered phase, one needs to include boson self interactions such that its energy is bounded from below.

companion work, Ref. [63]) that our results for these susceptibilities exactly agree with the direct calculation of the Drude weight.

Before concluding this Section, we present a simple derivation of the value of $M^2(\theta)$ using the $U(1)_{\text{patch}}$ anomaly. Consider introducing a probe, $\mu_\theta = \mu_{\theta+\pi}$, that couples only to two antipodal patches and varies slowly in the direction perpendicular to the Fermi surface, $\mu_\theta(\boldsymbol{w}(\theta) \cdot \boldsymbol{x})$. Then by the arguments of Appendix B, which assume a fully gauge invariant regulator, this will enter the anomaly equation as

$$q_\perp(\theta)\, \boldsymbol{w}(\theta) \cdot \langle \boldsymbol{j}_\theta(\omega = 0, q_\perp) \rangle_{\mu_\theta} = \frac{\Lambda_{\text{patch}}}{(2\pi)^2}\, q_\perp(\theta)\, \mu_\theta(\omega = 0, q_\perp) \,, \tag{3.51}$$

where we have fixed $\omega = 0$ and chosen the momentum to be in the direction perpendicular to the Fermi surface, $q_\perp(\theta) = \boldsymbol{w}(\theta) \cdot \boldsymbol{q}$. Using the relation, $\boldsymbol{w}(\theta) \cdot \boldsymbol{j}_\theta = v_F(\theta)\, n_\theta$, which is valid in the IR on dropping the curvature of the dispersion, one finds

$$\langle n_\theta(\omega = 0, q_\perp) \rangle_{\mu_\theta} = \frac{\Lambda_{\text{patch}}}{(2\pi)^2\, v_F(\theta)}\, \mu_\theta(\omega = 0, q_\perp) \,, \tag{3.52}$$

leading immediately to

$$\chi_{n_\theta n_{\theta'}} = \frac{\Lambda_{\text{patch}}}{(2\pi)^2\, v_F(\theta)}\, \delta_{\theta\theta'} \,. \tag{3.53}$$

We have therefore shown that the density susceptibility is fixed non-perturbatively by the anomaly and (equivalently) gauge invariance. Consequently, because $\chi_{\psi_\theta^\dagger \psi_\theta, \psi_\theta^\dagger \psi_\theta} = 0$ in the regularization where $\Lambda_\perp \to \infty$ before $\Lambda_\omega \to \infty$, it is necessary to add the term in Eq. (3.46) and revise the definition of $n_\theta$ to Eq. (3.47).

## 3.7 Effect of BCS pairing

We now consider how introducing BCS pairing interactions affects the arguments presented above. Such terms involve attractive interactions between fermions at antipodal points on the Fermi surface. In terms of the microscopic fields,

$$V_{\text{BCS}} = -\sum_{\mathbf{k},\mathbf{k}'} V_{\mathbf{k},\mathbf{k}'}\, \psi_\mathbf{k}^\dagger \psi_{-\mathbf{k}}^\dagger \psi_{\mathbf{k}'} \psi_{-\mathbf{k}'} \,. \tag{3.54}$$

In systems with inversion symmetry, each patch at momentum $\mathbf{k}$, with label $\theta$, has an antipode at momentum $-\mathbf{k}$, with label $\theta + \pi$. In the mid-IR theory, therefore, the BCS pairing term will take the form,

$$V_{\text{BCS}} = -\sum_{\theta \neq \theta'} V_{\theta,\theta'}\, \psi_\theta^\dagger \psi_{\theta+\pi}^\dagger \psi_{\theta'} \psi_{\theta'+\pi} \,. \tag{3.55}$$

The BCS interaction scatters fermions from patches $\theta$ and $\theta'$ into $\theta + \pi$ and $\theta' + \pi$. The precise form of this operator within each patch will not be important for our arguments.

Using a perturbative renormalization group approach, Ref. [64] found that theories with an inversion-even order parameter, where $g^a(\theta) = g^a(\theta + \pi)$ (such as Ising-nematic), are unstable to pairing and run to a superconducting ground state in which $U(1)_{\text{patch}}$ is broken spontaneously. The physics of this is that such order parameters mediate attractive interactions in the BCS channel. On the other hand, theories with an inversion-odd order parameter, where $g^a(\theta) = -g^a(\theta + \pi)$ (such as gauge theories and theories of "loop" order parameters discussed above), lead to *repulsion* in the BCS channel and flow to a metallic fixed point with *finite BCS couplings*. For a different approach yielding the same qualitative physics, see Ref. [33].

We wish to understand how the conclusions of this Section are affected, if at all, at such a finite-BCS fixed point. The potential issue is that the BCS coupling explicitly breaks the $U(1)_{\text{patch}}$ symmetry that was the centerpiece of our arguments. However, we do not expect that the conclusions should be modified at the finite-BCS fixed point in the IR limit. This expectation is based on the fact that, in models restricted to a single pair of antipodal patches, the finite BCS interaction does not break any symmetries. As discussed in Section 3.5, in the IR limit the critical boson fluctuations only couple strongly to antipodal patches, so the boson self energy should remain a sum over contributions from antipodal pairs, which are still individually determined by the anomaly equation.

There are a few cases in which one can strengthen this argument so that it is not necessary to assume the patch decoupling. Although the BCS interaction breaks the full $U(1)_{\text{patch}}$ symmetry, it still preserves a subgroup, generated by the total charge and the charge difference $n_\theta - n_{\theta+\pi}$ in each pair of antipodal patches. This subgroup still satisfies an anomaly equation. It follows that, if the coupling constants $g^a(\theta)$ are such that the boson effectively couples like a gauge field for the $U(1)$ generated by the total charge (as in, e.g. HLR theory), there is still a version of the anomaly equation; namely,

$$\partial_t(n_\theta - n_{\theta+\pi}) + \boldsymbol{\nabla} \cdot (\boldsymbol{j}_\theta - \boldsymbol{j}_{\theta+\pi}) = -2\frac{\Lambda_{\text{patch}}}{(2\pi)^2}\frac{1}{v_F(\theta)}g^a(\theta)\partial_t\phi_a \,. \tag{3.56}$$

One can show that the arguments in the previous sections then proceed similarly to before using this version of the anomaly equation.

Finally, if the order parameter is *even* under inversion symmetry, then the theory is unstable to pairing. Nevertheless, we may hope that the full constraints we obtained in earlier parts of Section 3, which relied on setting the BCS interaction to zero, describe the normal critical metallic state out of which the pairing instability develops. If the superconductivity

happens at sufficiently low temperature, then the metal just above the superconducting transition may be sufficiently "close" to the critical metallic fixed point (in the sense of irrelevant operators having run to zero) that our statements about the low energy dynamics may be pertinent. In this regime, the BCS couplings stay small and, like in an ordinary Fermi liquid, perhaps do not affect properties like the optical transport.

# 4 Microscopic sources of frequency scaling

## 4.1 Fixed point frequency scaling and its corrections

Based on the constraints following from the $U(1)_{\text{patch}}$ anomaly, we have demonstrated that for the metallic quantum critical points described by the mid-IR effective theory in Eq. (3.6), the boson self energy is indepedent of frequency and is *exactly* given by the formula,

$$\Pi^{ab}(\omega, \boldsymbol{q} = 0) = \sum_{\theta} \frac{\Lambda_{\text{patch}}}{(2\pi)^2} \frac{g^a(\theta) g^b(\theta)}{v_F(\theta)} \,. \tag{4.1}$$

We have also found that the optical conductivity is of the Drude form at the IR fixed point,

$$\text{Re}\,\sigma_{xx}(\omega) = \mathcal{D}\,\delta(\omega)\,, \tag{4.2}$$

where the Drude weight $\mathcal{D}$, is given by the general formulae in Eqs. (3.35) – (3.36). An important consequence of these results is that any further frequency dependence in the boson self energy or the optical conductivity of the mid-IR theory *vanishes* in the IR limit. In other words, critical fluctuations at the ultimate IR fixed point do not generate any frequency-dependent conductivity.

The above conclusions hold *in the IR fixed point theory.* Of course, the *microscopic theory* could still have non-trivial frequency scaling. We therefore clarify the implications of our result. In general, a signature of quantum criticality is a *scale invariant* contribution to the boson propagator, i.e. when frequency $\omega$ and temperature $T$ are sufficiently small, one would have

$$D(\omega, T) = \frac{1}{T^\gamma} \Gamma\left(\frac{\omega}{T}\right) + \cdots, \tag{4.3}$$

for some scaling exponent $\gamma$ and scaling function $\Gamma$. The scale invariance of this term shows that it must arise from the IR fixed point. Thus, one can interpret our result as proving that the only possibility is that $\gamma = 0$ and the scaling function $\Gamma$ is just a constant function. Similarly, for the optical conductivity what we have shown is that for a scale-invariant form of the optical conductivity,

$$\sigma(\omega, T) = \frac{1}{T^\alpha} \Sigma\left(\frac{\omega}{T}\right) + \cdots, \tag{4.4}$$

the only possibility is that $\alpha = 1$ and $\Sigma(x)$ is a constant times $i/x$.

There are standard caveats around such statements. In general, one can imagine expanding the microscopic field $\phi$ in terms of the scaling fields $\mathcal{O}_i$ in the IR,

$$\phi_{\text{microscopic}} = \phi + c_1 \mathcal{O}_1 + c_2 \mathcal{O}_2 + \cdots , \tag{4.5}$$

where $c_1, c_2, \ldots$ are dimensionful constants and the expansion is in operators of increasing scaling dimension. The boson propagator that we have fixed from general arguments corresponds to the *leading* operator in the expansion, $\phi$, which has the smallest scaling dimension. If one calculates the propagator of the microscopic boson field at the IR fixed point, there could be additional contributions from the sub-leading operators in the expansion. Because the calculation is performed in the IR fixed point theory, these contributions would still have a universal, scale invariant form. In general, the scaling exponent and scaling function associated with such "secondary" fixed-point scaling cannot be constrained by our general arguments. There are analogous caveats for the current correlator that we will discuss later in this Section.

At zero temperature, it is also possible for *corrections to scaling*, $\sim \omega^\lambda$, to arise in the optical conductivity or the boson propagator due to the inclusion of irrelevant operators in the Hamiltonian, thus going beyond the fixed-point theory. Moreover, the exponent $\lambda$ can still be universal, as it would be determined by the scaling exponent of the leading irrelevant operator that contributes. In contrast to the "primary" and "secondary" fixed point scaling discussed above, there is no reason for such corrections to scaling to obey a scale-invariant form like Eq. (4.3) at finite temperature.

## 4.2 Taxonomy of irrelevant operators

Beyond the general statements made above, we can use symmetry and anomaly arguments to draw concrete conclusions about which irrelevant operators could contribute to frequency scaling in the mid-IR theory, Eq. (3.5). In particular, recall that the only assumptions required for the derivation of absence of frequency dependence in the boson self-energy are that the boson action is quadratic in the mid-IR theory, and that the $U(1)_{\text{patch}}$ anomaly equation holds, governing the response of the fermions to the boson. It follows that the boson self-energy can only acquire a frequency dependence from adding the following classes

of irrelevant operators to the Hamiltonian[11]

- Boson self-interactions such as $\phi^4$. These operators lead to additional 1PI contributions to the boson propagator that are not controlled by the anomaly.

- Operators that explicitly break the global $U(1)_{\text{patch}}$ symmetry. Examples include operators that scatter a fermion from one patch into another.

- Operators that modify the coupling to the boson or alter the fermion Lagrangian such that the boson no longer couples as a $U(1)_{\text{patch}}$ gauge field. Simple examples include adding the curvature of the dispersion, $\kappa_\perp \psi_\theta^\dagger (\boldsymbol{w}(\theta) \cdot \boldsymbol{\nabla})^2 \psi_\theta$ (with ordinary, rather than covariant, derivatives), or introducing intra-patch momentum dependence to the boson-fermion coupling, $g^a(\theta; \mathbf{k})$.

The last two categories of interactions would invalidate the $U(1)_{\text{patch}}$ anomaly equation. All three categories of interactions were explicitly excluded in our definition of the mid-IR theory.

For the conductivity, the situation is somewhat more subtle. As discussed in Section 3.5, one can write the current operator as a sum,

$$\boldsymbol{J} = \boldsymbol{J}_\perp^{(0)} + \boldsymbol{J}_\parallel^{(0)} + \cdots \tag{4.6}$$

Here $\boldsymbol{J}_\perp^{(0)}$ and $\boldsymbol{J}_\parallel^{(0)}$ are the components of the current operator from directions parallel and perpendicular to the Fermi surface that come from minimally coupling the Hamiltonian of the mid-IR theory to a vector potential, $\boldsymbol{A}$. The "$\cdots$" contains contributions to the current operatator arising from minimally coupling to irrelevant operators not included in the mid-IR theory, as well as from non-minimal couplings. Simple examples come from higher-derivative terms in the expansion of the dispersion, e.g. $\kappa^{(n)} \psi_\theta^\dagger (\boldsymbol{w}(\theta) \cdot \boldsymbol{\nabla})^n \psi_\theta$, which lead to corrections to the definitoin of $\boldsymbol{J}_\perp$.

One can think of Eq. (4.6) as a special case of the general expansion of a microscopic operator in terms of IR fields mentioned earlier. With a notable exception that we will discuss in the paragraph below, we do not have constraints on the contributions of the conductivity coming from the terms in the "$\cdots$" in Eq. (4.6), corresponding to what we referred to above as "secondary" fixed point scaling.

---

[11] Here our focus is on correlators of the boson field as defined in the mid-IR theory, which will not be affected by additional "secondary" fixed point scaling. However, we caution that correlators of the boson field appearing in microscopic models like those considered in numerics *will* be affected by such secondary scaling, as in Eq. (4.5).

Importantly, we showed in Section 3.5 that the contribution from the first two terms in Eq. (4.6) in the IR fixed point theory are constrained. Furthermore, the two-point correlation function of $\boldsymbol{J}_{\perp}^{(0)}$ was constrained under the same assumptions as the boson propagator; hence it may only receive corrections to frequency scaling from the three classes of irrelevant operators described above that lead to corrections to scaling in the boson propagator. However, the vanishing of the $\boldsymbol{J}_{\parallel}^{(0)}$ correlator relied on emergent intra-patch momentum conservation at the IR fixed point. Therefore, *any* irrelevant operator that leads to exchange of momentum between pairs of (non-antipodal) patches could affect its frequency scaling. Finally, let us remark that the contribution from the part of the current operator that comes from minimally coupling to the quadratic term in the expansion of the dispersion in the direction perpendicular to the Fermi surface, i.e. $\kappa^{(2)}\psi_\theta^\dagger(\boldsymbol{w}(\theta)\cdot\boldsymbol{\nabla})^2\psi_\theta$, a term which was *not* included in the mid-IR theory, actually does vanish in the fixed-point theory for the same reason as for $\boldsymbol{J}_{\parallel}^{(0)}$.

# 5  Revisiting claims of universal frequency scaling

The results described above appear in tension with several earlier calculations of the boson self energy and/or optical conductivity in related theories over the years [27, 37, 54, 55]. These works have suggested that there is a non-trivial universal frequency scaling. In particular, the results for the self energy are of the general form,

$$\Pi(\omega, \boldsymbol{q} = 0) = C_0 + C_z\,\mathrm{sgn}(\omega)|\omega|^{1-p(z)} + \dots, \tag{5.1}$$

where $C_0, C_z$ are dimensionful constants, $p(z)$ depends on the dynamical critical exponent of the bosons, and the ellipses denote less singular terms. Although different approaches have found different forms of $p(z)$, they all agree that $p(z = 3) = 2/3$. For $C_z \neq 0$, these results apparently contrast with our conclusion that the boson self-energy is independent of frequency. Although we described in Section 4 some mechanisms for universal frequency scaling due to irrelevant operators, it is not immediately clear that this is actually what is behind the results found in the past works, which we revisit in this Section.

Below, we will discuss the two main techniques leading to results of the form in Eq. (5.1): (1) Migdal-Eliashberg theory, in which the self energy is obtained by postulating a solution to a set of self-consistent equations, and (2) the fundamental large $N$ expansion, the $N \to \infty$ limit of which corresponds to the random phase approximation (RPA), which was famously used in Ref. [27] to obtain $p(z = 3) = 2/3$ in the context of a Fermi surface coupled to a gauge field. We will find below that Migdal-Eliashberg theory leads to results that are incon-

sistent with our general arguments based on the anomaly and therefore should be considered invalidated in the present context. In contrast, we will show that applying the fundamental large $N$ expansion to the mid-IR theory in fact yields $C_z = 0$ for arbitrary fermion dispersion and form factor symmetry. It is thus consistent with our general arguments.

We emphasize that both of the above calculations invoke uncontrolled approximations. We will discuss the appearance of the anomaly in controlled treatments of the problem in Section 6.

## 5.1   Migdal-Eliashberg theory

For general models of fermions at finite density coupled to bosons, the Migdal-Eliashberg approach involves approximately solving the exact Schwinger-Dyson equations by including all self energy corrections but neglecting vertex corrections. In the context of the electron-phonon problem, for example, this approximation was justified by the Migdal's theorem, which states that the vertex corrections are suppressed by powers of $\frac{m_*}{M}$, where $m_*$ is the effective mass and $M$ is the ion mass. In models of Fermi surfaces coupled to critical bosons, as in Eq. (3.1), Migdal-Eliashberg theory provides a set of self-consistent equations for the fermion self energy, $\Sigma$, and boson self energy, $\Pi$,

$$\Pi(\Omega, \boldsymbol{q}) = \int_{\boldsymbol{k}, \omega} f(\boldsymbol{k}, \boldsymbol{q})^2 \, G(\omega, \boldsymbol{k}) \, G(\omega + \Omega, \boldsymbol{k} + \boldsymbol{q}) \,, \tag{5.2}$$

$$\Sigma(\omega, \boldsymbol{k}) = \int_{\boldsymbol{q}, \Omega} f(\boldsymbol{k}, \boldsymbol{q})^2 \, G(\omega + \Omega, \boldsymbol{k} + \boldsymbol{q}) \, D(\Omega, \boldsymbol{q}) \,, \tag{5.3}$$

where $f(\boldsymbol{k}, \boldsymbol{q})$ are model-dependent vertex factors, and $\int_{\boldsymbol{k}, \omega} \equiv \int \frac{d^2\boldsymbol{k} d\omega}{(2\pi)^3}$. In the low frequency limit, these equations can be solved exactly assuming a dynamical critical exponent for the boson fluctuations, $z$. One then obtains a solution for the boson self energy, $\Pi(\Omega, \boldsymbol{q} = 0) = C_0 + C_z \operatorname{sgn}(\Omega) |\Omega|^{1 - \frac{2}{z}}$, $C \neq 0$ (see Ref. [37] for a brief review of this calculation). The physical values of $z$ usually fall in the range $2 < z \leq 3$. Therefore, Eliashberg theory predicts a nontrivial frequency dependence of the self energy. One can verify that this remains true even when computing directly in the mid-IR theory [63]. If the standard Eliashberg theory were exact, then these results would contradict the constraints from the anomaly.

We do not find this disagreement surprising, as the Migdal-Eliashberg approximation is an uncontrolled truncation of the full set of Feynman diagrams that artificially favors fermion self energy diagrams over vertex corrections. Interestingly, an augmented version of Eliashberg theory proposed in Ref. [37] does give results consistent with the anomaly. In this approach, the fermion self energy, $\Sigma(\omega, \boldsymbol{k})$, remains approximately independent of $|\boldsymbol{k}|$, while

a subset of the vertex corrections beyond the standard Eliashberg theory are resummed more systematically. This procedure leads to a result for $\Pi(\Omega, \boldsymbol{q} = 0)$ in which nontrivial frequency scaling is entirely generated by irrelevant operators that in our mid-IR theory correspond to the momentum dependence of the boson-fermion coupling, $g^a(\theta; \boldsymbol{k})$, within each patch. These results are consistent with Ward identities developed in Ref. [37], which for the models they consider complement our approach leveraging anomalies.

More recently, it has been shown that although the original Migdal-Eliashberg self-consistent equations cannot be derived from the conventional model of Fermi surfaces coupled to critical bosons, they arise as the saddle point solution to a large-$N$ model with random Yukawa couplings, which was inspired by the Sachdev-Ye-Kitaev (SYK) model [35, 36, 65]. As we will see in Section 6, that model has a more complex anomaly structure that does not fix the boson self energy completely as in the single-flavor model considered in Section 3. We present a detailed calculation of the optical conductivity in models of that type in a companion paper, focusing on "loop current" order parameters [63].

## 5.2 RPA and fundamental large $N$

The detailed calculation of the boson self energy within the random phase approximation (RPA) first appeared in Ref. [27] for a theory of a Fermi surface coupled to a fluctuating gauge field in two spatial dimensions, building on earlier work on the half-filled Landau level [23]. The authors of Ref. [27] studied the following Euclidean action describing $N_f$ fermion fields, $\psi_I$, $I = 1, \ldots, N_f$, coupled to a vector boson, $\vec{\phi} = (\phi_x, \phi_y)$,

$$S = S_\psi + S_{\text{int}} + S_\phi \,, \tag{5.4}$$

$$S_\psi = \int_{\omega, \boldsymbol{k}} \psi_I^\dagger(\boldsymbol{k}, i\omega)[i\omega - \epsilon(\boldsymbol{k})]\psi_I(\boldsymbol{k}, i\omega) \,, \tag{5.5}$$

$$S_{\text{int}} = \int_{\Omega, \boldsymbol{q}} \int_{\omega, \boldsymbol{k}} \vec{\phi}(\boldsymbol{q}, i\Omega) \cdot \vec{v}(\boldsymbol{k}) \, \psi_I^\dagger(\boldsymbol{k} + \boldsymbol{q}, i\omega + i\Omega)\psi_I(\boldsymbol{k}, i\omega) \,, \tag{5.6}$$

$$S_\phi = \frac{N_f}{2\alpha} \int_{\Omega, \boldsymbol{q}} [\boldsymbol{q} \times \vec{\phi}(-\boldsymbol{q}, -i\Omega)] \frac{1}{|\boldsymbol{q}|^{3-z}} [\boldsymbol{q} \times \vec{\phi}(\boldsymbol{q}, i\Omega)] \,, \tag{5.7}$$

where again $\int_{\omega, \boldsymbol{k}} \equiv \int \frac{d^2\boldsymbol{k}\, d\omega}{(2\pi)^3}$ and $\vec{v}(\boldsymbol{k}) = \partial_{\boldsymbol{k}}\epsilon(\boldsymbol{k})$. Here $z > 2$ is the dynamical critical exponent of the boson. Although $\vec{\phi}$ appears here as a gauge field, a more general situation can be considered by replacing $\vec{v}(\boldsymbol{k})$ with some general form factor, $\vec{g}(\boldsymbol{k})$.

The bare propagators for the fermion and the transverse component of $\vec{\phi}$, $\phi_{\text{T}}(\boldsymbol{k}, i\omega) \equiv$

$\varepsilon_{ij}k_i\phi_j(\boldsymbol{k}, i\omega)/|\boldsymbol{k}|$, are

$$G_{IJ}(\boldsymbol{k}, i\omega) = \frac{\delta_{IJ}}{i\omega - \epsilon(\boldsymbol{k})} \quad , \quad D_0(\boldsymbol{q}, i\Omega) = \frac{\alpha}{N_f} \frac{1}{|\boldsymbol{q}|^{z-1}} . \tag{5.8}$$

The random phase approximation (RPA) corresponds to taking $N_f \to \infty$ while holding $\alpha$ fixed. In this limit, the fermion self energy, $\Sigma(\boldsymbol{k}, i\omega)$ is suppressed, while the boson self energy, which takes the Landau damping form, $\Pi_{\mathrm{TT}}(\boldsymbol{q}, i\Omega) = \gamma_{\boldsymbol{q}} \frac{|\Omega|}{|\boldsymbol{q}|}$ (in Coulomb gauge, $\vec{\nabla} \cdot \vec{\phi} = 0$) is summed in a geometric series, yielding

$$D(\boldsymbol{q}, i\Omega) = \frac{\alpha}{N_f} \frac{1}{|\boldsymbol{q}|^{z-1} + \gamma_{\boldsymbol{q}} \frac{|\Omega|}{|\boldsymbol{q}|}} . \tag{5.9}$$

Using $G_{IJ}$ and $D$ in internal loops, one can then compute correlation functions in a diagrammatic expansion in powers of $1/N_f$.

For the special case of a rotationally invariant Fermi surface coupled to a $U(1)$ gauge field, Ref. [27] found that the boson self energy at $\mathcal{O}(1/N_f)$ takes the scaling form in Eq. (5.1) with

$$p_{\mathrm{RPA}}(z) = \frac{2z - 4}{z} \tag{5.10}$$

and $C_z$ left undetermined. For $z = 3$, the same frequency scaling was found in Fermi surfaces coupled to gapless nematic order parameters, up to additional corrections due to disorder and Umklapp scattering [37–39, 66]. Despite the fact that this large-$N_f$ expansion has been demonstrated to be uncontrolled in the low frequency limit[12] [29], the result $p(z) = \frac{2z-4}{z}$ is often cited as evidence of fixed point frequency scaling in the self energy as well as the conductivity.

Here we apply the large-$N_f$ approach of Ref. [27] to the context of our mid-IR effective theory, rather than the full microscopic theory in Eq. (5.4), and we demonstrate the *absence* of any frequency scaling in the boson self energy (i.e. $C_z = 0$) to $\mathcal{O}(1/N_f)$. We furthermore find that this result extends to general choices of order parameter symmetry and fermion dispersion. Hence, despite the fact that this expansion is uncontrolled, we are tempted to regard the origin of the $|\Omega|^{\frac{4-z}{z}}$ scaling found in Ref. [27] as irrelevant operators that invalidate the $U(1)_{\mathrm{patch}}$ anomaly, as discussed above.

---

[12]This can be understood by noting that the fermion self energy is $\Sigma(\omega) \sim \mathrm{sgn}(\omega)|\omega|^{2/z}/N_f$. For $z > 2$, this term dominates over the linear in frequency term in the fermion propagator at very low frequencies. In vertex corrections, this leads to an enhancement of virtual fermion propagators by powers of $N_f$, rendering the expansion uncontrolled unless the $N_f \to \infty$ limit is taken prior to the low frequency limit.

We now summarize the large-$N_f$ computation of the boson self energy, $\Pi(\boldsymbol{q} = 0, i\Omega)$, in the mid-IR theory, leaving the detailed calculations to Appendix C. Extending the mid-IR theory to include $I = 1, \ldots, N_f$ fermion species, we consider the action,

$$S = S_{\text{boson}} + \sum_\theta S_{\text{patch}}(\theta) \,, \tag{5.11}$$

where

$$S_{\text{patch}}(\theta) = \int_{\tau,\boldsymbol{x}} \left[ \psi_{I,\theta}^\dagger \left\{ \partial_\tau + iv_F(\theta)[\boldsymbol{w}(\theta) \cdot \boldsymbol{\nabla}] + \kappa_{ij}(\theta)\partial_i\partial_j \right\} \psi_{I,\theta} + \frac{g(\theta)}{\sqrt{N_f}} \phi \, \psi_{I,\theta}^\dagger \psi_{I,\theta} \right], \tag{5.12}$$

$$S_{\text{boson}} = \int_{\tau,\boldsymbol{q}} \frac{1}{2} \phi(-\boldsymbol{q}, \tau) \left[ -\lambda \, \partial_\tau^2 + |\boldsymbol{q}|^{z-1} \right] \phi(\boldsymbol{q}, \tau) \,. \tag{5.13}$$

Note that we have specialized to the case of $N_b = 1$ boson components and rescaled $\phi$ by $\sqrt{N_f}$ (for the case of a U(1) gauge field, this action corresponds to choosing Coulomb gauge). As in the discussion in Section 3, different order parameter symmetries correspond to different choices of $g(\theta)$.

The six diagrams that that contribute to $\mathcal{O}(1/N_f)$ correspond to those shown in Figure 2. In each diagram, the solid lines are bare fermion propagators, and the wavy lines are the Landau damped boson propagators, as in Eq. (5.9). In the RPA limit ($N_f \to \infty$), the only contribution to $\Pi(\boldsymbol{q} = 0, i\omega)$ comes from the one-loop bubble diagram. For a general form factor, $g(\theta)$, and a general dispersion, $\epsilon(\boldsymbol{k})$, this diagram alone reproduces the anomaly prediction in Eq. (3.18). The $1/N_f$-corrections to the RPA result come from fermion self energy and vertex corrections, as well as the so-called Aslamazov-Larkin diagrams that capture scattering of electrons off of boson pairs (the two diagrams at the bottom of Figure 2).

The self energy and vertex corrections are constrained by a Ward identity, and we find that they cancel exactly in the mid-IR theory. Any putative frequency dependence must then come from the Aslamazov-Larkin diagrams. Remarkably, it turns out that these diagrams cancel among themselves as well, as shown in Appendix C. Therefore, to this order, the fundamental large-$N_f$ expansion is fully consistent with the U(1)$_{\text{patch}}$ anomaly within the mid-IR theory.

How, then, should we interpret the nontrivial frequency scaling $C_z|\Omega|^{\frac{4-z}{z}}$ found in Ref. [27]? A direct comparison of the results in Ref. [27] with our mid-IR calculations is not justified because Ref. [27] worked with the full "microscopic" theory of non-relativistic fermions, and so their calculation involves operators that were thrown out en route to the mid-IR theory. We provide a detailed analysis of the effects of irrelevant operators in the calculation of Ref. [27] in a companion paper [63].

# 6  Applicability of the anomaly constraints to controlled expansions

Thus far, we have shown that the $\text{U}(1)_{\text{patch}}$ anomaly provides non-perturbative constraints on theories of Fermi surfaces coupled to critical bosons, as in Eq. (3.1). Such constraints can complement attempts to develop controlled techniques for capturing the IR dynamics of these theories. Because the class of theories described by Eq. (3.1) does not involve any small parameter with which one can perform perturbation theory, any such technique involves deforming the theory to introduce a small parameter, such that the original theory is recovered when the parameter is order one. One then studies the new theory that results, with the hope that continuing the small parameter to one can capture the physics of the original theory. Any perturbative scheme hoping to capture the full IR physics of the physical fixed point ideally ought to satisfy the non-perturbative constraints we derived in Section 3 on the boson propagator and optical conductivity for the original theory. In this section, we discuss the extent to which this statement holds in various deformations of Eq. (3.1) by considering how the anomaly manifests in each. In particular, we focus on cases that are believed to lead to controlled perturbative expansions.

## 6.1  Fundamental large $N$ with small $z - 2$

After Ref. [29] demonstrated the breakdown of RPA, there was an immediate effort to seek new controlled expansions. One of the earliest proposals [30] combined the RPA (or large-$N_f$) approach described in Section 5.2 with an expansion in the dynamical exponent of the boson, $z = 2 + \epsilon$, which was originally used by Nayak and Wilczek to develop a renormalization group flow [26]. By taking the limit of small $\epsilon$ and large number of fermion species, $N_f$, holding the product $N_f \epsilon$ fixed, Ref. [30] argued that many of the correlation functions of interest in the quantum critical regime $\omega \ll v_F |\boldsymbol{q}|$ are organized into a systematic expansion in powers of $1/N_f$ (or equivalently $\epsilon$). As discussed in Section 5.2, introducing large $N_f$ and allowing $z$ to vary does not alter the anomaly except to multiply it by $N_f$. Therefore, we expect $\Pi(\omega, \boldsymbol{q} = 0)$ to be independent of $\omega$ order by order in the large $N_f$ expansion.

The validity of this expansion was recently called into question in Ref. [31], who found $\log^2$ divergences in a subset of three-loop diagrams due to large-momentum scattering in the $\epsilon = 0$ marginal Fermi liquid base theory that causes virtual Cooper pairs to spread across the entire Fermi surface. Nevertheless, since this divergence appears only at third leading order in $1/N_f$, one can still hope to verify the prediction of the anomaly to leading order in

$1/N_f$. Unfortunately, although the fermion self energy $\Sigma(\omega, \boldsymbol{k})$ only receives contributions from a finite set of diagrams at each order, the boson self energy $\Pi(\Omega, \boldsymbol{q} = 0)$ sums an infinite number of diagrams that are difficult to compute even at leading order in large $N_f$.

To make this observation more precise, we evaluate the diagrams in Figure 2 for general $z = 2 + \epsilon$ and take the limit as $\epsilon \to 0$ in the end. Following Section 5.2, for any $\epsilon > 0$, we can associate a factor of $N_f$ to each fermion loop and a factor of $\sqrt{1/N_f}$ to each boson-fermion vertex. Applying this general counting scheme to the diagrams in Figure 2, we find that the one-loop bubble diagram is $\mathcal{O}(1)$ and the remaining five diagrams are all formally $\mathcal{O}(1/N_f)$. However, as we take the limit $\epsilon \to 0$, the coefficients of the $1/N_f$-corrections to the boson self energy at $\boldsymbol{q} = 0$ diverge as

$$\Pi_{1/N_f}(\omega, \boldsymbol{q} = 0) \sim \frac{\pi(\omega)}{N_f \epsilon}, \tag{6.1}$$

where $\pi(\omega)$ is a regular function of $\omega$. Since the limit $\epsilon \to 0$ is taken with $N_f \epsilon$ fixed, the above scaling implies that formally $\mathcal{O}(1/N_f)$ contributions to $\Pi(\omega, \boldsymbol{q} = 0)$ are enhanced by a power of $1/\epsilon \sim N_f$. Therefore, the naïve $1/N_f$ counting breaks down.

The origin of this breakdown is the divergence of internal boson momentum integrals in the $\epsilon \to 0$ limit. To see this, we recall the RPA boson propagator in Section 5.2

$$D(\boldsymbol{q}, i\Omega) = \frac{1}{|\boldsymbol{q}|^{1+\epsilon} + \gamma_{\boldsymbol{q}} \frac{|\Omega|}{|\boldsymbol{q}|}}. \tag{6.2}$$

For all $\epsilon > 0$, the integral of $D(\boldsymbol{q}, i\Omega)$ over $|\boldsymbol{q}|$ converges. However, the prefactor of the integral has an $\epsilon^{-1}$ divergence as $\epsilon \to 0$. This general structure allows an infinite number of diagrams to contribute to leading order in the $1/N_f$ expansion. For example, in a ladder diagram with $n$ rungs, each rung adds two vertices, one boson propagator, and no fermion loop. By naïve counting, such a diagram should be suppressed by $N_f^{-n}$ relative to the one-loop bubble. But the above discussion implies that each rung also brings an additional integral over an internal boson propagator. The $\epsilon^{-1}$ divergence in the integral compensates for the $N_f^{-n}$ suppression, making ladder diagrams with any number of rungs contribute to the same order as the one-loop bubble. It would be interesting to systematically classify all diagrams that contribute to leading order in the $1/N_f$ expansion and explicitly verify the anomaly. We leave this to future work.

## 6.2 Co-dimensional regularization

A rather different kind of expansion was introduced by Dalidovich and Lee [32]. These authors constructed a dimensional regularization scheme that continues the Fermi surface

codimension from 1 to $1.5 - \epsilon$, while keeping the bare boson dynamical critical exponent fixed. Since the Fermi surface itself always has dimension fixed to 1, a boson with small momentum, $|\boldsymbol{q}| \ll k_F$, still couples most strongly to anti-podal patches on the Fermi surface, and the critical properties of the theory are controlled by a quantum field theory localized on these patches. Combining fermions in the two patches into a Euclidean Dirac spinor $\Psi_I(k) = (\psi_{+,I}(k), \psi^\dagger_{-,I}(k))$, $I = 1, \ldots, N_f$, the two-patch action for general co-dimension $d - 1$ can be written as

$$S_{\text{patch}} = S_\Psi + S_{\text{int}} + S_\phi \,, \tag{6.3}$$

$$S_\psi = \int_{\omega,\boldsymbol{k}} \bar{\Psi}_I [i \boldsymbol{\Gamma} \cdot \mathbf{K} - i \gamma_{d-1} \epsilon(k_{d-1}, k_d)] \Psi_I \,, \tag{6.4}$$

$$S_{\text{int}} = \sqrt{\frac{d-1}{N_f}} \int_{\omega,\boldsymbol{k}} \int_{\Omega,\boldsymbol{q}} \phi(\boldsymbol{q}, \Omega) \bar{\Psi}_I(\boldsymbol{k} + \boldsymbol{q}, \omega + \Omega) \, M \, \Psi_I(\boldsymbol{k}, \omega) \,, \tag{6.5}$$

$$S_\phi = \frac{1}{2\alpha} \int_{\Omega,\boldsymbol{q}} \phi(-\boldsymbol{q}, -\Omega) \, (\Omega^2 + |\boldsymbol{q}|^2) \, \phi(\boldsymbol{q}, \Omega) \,, \tag{6.6}$$

where we have restricted to a single boson species, $\phi$, chosen $k_{d-1}, k_d$ to denote the two dimensional momentum space of the physical theory, defined $\mathbf{K} = (k_0, k_1, \ldots, k_{d-2}, 0, 0)$ to include the frequency and the momenta of the ambient space, and defined $\boldsymbol{\Gamma}$ to be the corresponding Euclidean Dirac $\gamma$ matrices. We also define $M$ to be some product of gamma matrices depending on the symmetry properties of the order parameter. By expanding in the co-dimension, $d - 1 = 3/2 - \epsilon$, it is possible to construct a controlled IR fixed point at which the boson has self energy [32, 54, 55],

$$\Pi(\boldsymbol{q} = 0, \Omega) \sim |\Omega|^{1/z} \,, \tag{6.7}$$

where $z$ is the dynamical critical exponent. This generally leads to a frequency-dependent conductivity as in Eq. (5.1), with $p(z) = (1 - z)/z$. We emphasize that in this family of models, $z$ is tuned by varying the codimension $\epsilon$ while keeping the boson kinetic term local (i.e. the bare propagator is an analytic function of momenta).

The reason non-trivial frequency scaling is permitted in the dimensionally continued theory is that the $\mathrm{U}(1)_{\text{patch}}$ symmetry and, by extension, the global charge conservation symmetry are explicitly broken at the level of the action for general co-dimension ($\epsilon \neq 1/2$). Our general anomaly arguments therefore do not apply to such situations. This symmetry breaking appears because, for general $N_f$ and $d - 1$, the terms involving $\boldsymbol{\Gamma} \cdot \mathbf{K}$ in $S_\psi$ source Cooper pair operators. As a result, $\mathrm{U}(1)_{\text{patch}}$ is broken explicitly, and the anomaly is no longer meaningful. This suggests that while the co-dimension expansion may be useful for

extracting some universal properties of the theory at $\epsilon = 1/2$, it is not reliable for computing correlation functions that are constrained by the anomalous $U(1)_{\text{patch}}$ symmetry (i.e. through Ward identities). In particular, it is not meaningful to talk about the electrical conductivity, as the global charge conservation is explicitly broken.

## 6.3 Matrix large $N$

The deformation considered in Refs. [28, 33, 34] again involves $N_c$ species of fermions in the fundamental representation, but now the bosonic degrees of freedom are promoted to a matrices $\phi_{IJ}$ transforming in the adjoint representation of $U(N_c)$ [13]. The matrix field couples to the fermions in a patch $\theta$, $\psi_\theta$, like a $U(N_c)$ gauge field,

$$S_{\text{int}} = \int_{\tau,\boldsymbol{x}} \sum_\theta g(\theta) \psi_{\theta,I}^\dagger \phi_{IJ} \psi_{\theta,J} \,. \tag{6.8}$$

Near the critical point, the boson action contains a $U(N_c)$ invariant mass term, $m^2 \operatorname{Tr} \phi^2$, and the critical point at $r = 0$ can be accessed by tuning a single parameter in the UV. In analogy with the theory considered in Section 3, the present theory enjoys a $U(N_c)_{\text{patch}}$ symmetry. Generalizing the arguments in Section 3 and expanding the boson and current operators in the basis of generators of $U(N_c)$,

$$\phi_{IJ} = \frac{1}{N_c} \operatorname{Tr}[\phi] \, \delta_{IJ} + \phi_a t_{IJ}^a \,, \tag{6.9}$$

where $t_{IJ}^a$ are the generators of $SU(N_c)$, we obtain a non-Abelian version of the anomaly equation,

$$\partial_\mu \operatorname{Tr}[J_\theta^\mu] = -\frac{\Lambda_{\text{patch}}}{(2\pi)^2} \frac{g(\theta)}{v_F(\theta)} \partial_t \operatorname{Tr}[\phi] \tag{6.10}$$

$$\partial_t n_\theta^a + D_i J_\theta^{i,a} = -\frac{\Lambda_{\text{patch}}}{(2\pi)^2} \frac{g(\theta)}{v_F(\theta)} \partial_t \phi^a \,, \quad D_i J_\theta^{i,a} = \partial_i J_\theta^{i,a} + i f^{ab}{}_c \phi_b J_\theta^{i,c} \,, \tag{6.11}$$

where $f^{abc}$ are the structure constants of $SU(N_c)$ and $(J_\theta^\mu)_{IJ} = ((n_\theta)_{IJ}, (\boldsymbol{J}_\theta)_{IJ})$ is the $U(N_c)_{\text{patch}}$ current, which we have written above in terms of its trace and non-Abelian components, $J_\theta^{\mu,a}$, which couple to $\phi^a$. Note that both sides of the anomaly equations transform in the adjoint representation of $U(N_c)$.

---

[13]Refs. [28, 33, 34, 67, 68] actually consider the case of a $SU(N_c)$ boson for simplicity, but their approach can be equivalently applied to $U(N_c)$ bosons. We also note that the theories considered in those works do not involve a patch decomposition, so we must make the (we think reasonable) assumption that the mid-IR patch theory flows to the same IR fixed point as those theories.

Importantly, the covariant derivative that appears on the left-hand side of Eq. (6.11) implies that the anomaly equation mixes density and current even at $\boldsymbol{q} = 0$, so the SU($N_c$) components of the current operator at $\boldsymbol{q} = 0$ are not longer simply related to the bosonic field $\phi$, as was the case for the Abelian currents considered in Section 3. However, the physical EM current in this model is the trace, Tr[$J$], which satisfies the Abelian anomaly equation, Eq. (6.10). Therefore, for this current we can obtain

$$\frac{d}{dt} \operatorname{Tr} J(\boldsymbol{q} = 0, t) = -\frac{\Lambda_{\text{patch}}}{(2\pi)^2} \frac{g(\theta)}{v_F(\theta)} \frac{d}{dt} \operatorname{Tr} \phi(\boldsymbol{q} = 0, t) \,, \qquad (6.12)$$

which, by the same arguments as in Section 3, is enough to allow us to fix the propagator of Tr[$\phi$] and therefore the conductivity to the results in Eq. (3.19) and (3.35). However, this equation cannot fully constrain the SU($N_c$) components of the boson self energy, $\Pi_{ab}(q = 0, \omega)$, meaning that we cannot determine the propagator for the SU($N_c$) components of $\phi$.

## 6.4  Random-flavor large $N$

Finally, we consider the most recently developed expansion, which we refer to as the "random-flavor large $N$ expansion" [35, 36, 65]. This approach was inspired by the SYK model, and it leverages quenched randomness in the space of fermion and boson species to construct a theory for which the Eliashberg solution to the Schwinger-Dyson equations appears as a $N \to \infty$ saddle point. As in the matrix large-$N$ example, one considers $N_f = N$ flavors of fermions, $\psi_I$, but now each couples to a boson which is a $N_b = N$ component *vector*, $\phi_I^a$, via random trilinear couplings, $g_{IJK}$,

$$S = S_\psi + S_\phi + \int \sum_\theta \frac{g_{IJK} f(\theta)}{N} \psi_{\theta,I}^\dagger \psi_{\theta,J} \phi_K \,, \qquad (6.13)$$

where $f(\theta)$ is a form factor determined by the symmetry of the order parameter. One could also consider order parameters for which the form factor carries an additional index, $f^a(\theta)$ as we did in Section 3, but we suppress it here. We choose the couplings $g_{IJK} = g_{JIK}^*$ (couplings not satisfying this property can also be considered), to be a set of independent complex Gaussian random numbers with zero mean,

$$\overline{g_{IJK}} = 0 \,, \overline{g_{IJK} g_{I'J'K'}^*} = g^2 \, \delta_{II'} \delta_{JJ'} \delta_{KK'} \,. \qquad (6.14)$$

In the $N \to \infty$ limit, the theory self-averages so that quenched and annealed averages match.

After performing the annealed average, the partition function of the original $2N$ fields can be recast into a path integral with four bilocal fields and an action that's $\mathcal{O}(N)$. The

saddle point equation of this path integral precisely corresponds to the standard Eliashberg equations. Corrections to the Eliashberg theory can then be organized into a $1/N$ expansion, given that the assumption of self-averaging does not break down. Moreover, at large but finite $N$ each boson species should be independently tuned to criticality, as the ensemble is not actually invariant under $\mathrm{O}(N)$. Hence, there are $N^2$ independent mass bilinears, $R_{IJ}\phi_I\phi_J$, allowed by symmetry in each disorder realization that need to be tuned to criticality, meaning that the $N \to \infty$ saddle becomes a multicritical point that may be in a distinct universlity class from the fixed points of the theories described in the above subsections. We comment more extensively on these issues in a companion paper [63].

The random-flavor model has a similar anomaly structure to the matrix large-$N$ theory, since in each realization fluctuations of $g_{IJK}f(\theta)\phi_K$ are almost those of a $\mathrm{U}(N)$ gauge field, but with a constraint that restricts its allowed configurations. Thus, the anomaly equations are those in Eqs. (6.10) – (6.11) with $g\phi_{IJ}$ replaced with $g_{IJK}f(\theta)\phi_K$. In particular, the analogue of Eq. (6.10) is

$$\partial_t n_\theta + \boldsymbol{\nabla} \cdot \boldsymbol{j}_\theta = -\frac{\Lambda_{\text{patch}}}{(2\pi)^2}\frac{1}{v_F(\theta)}\sum_I g_{IIK}f(\theta)\partial_t\phi_K . \tag{6.15}$$

The anomaly for the off-diagonal components of $g_{IJK}\phi_K$, like the $\mathrm{SU}(N)$ anomaly equation in Eq. (6.11), similarly involves covariant derivatives and mixes density and current even at $\boldsymbol{q} = 0$. As in the matrix large-$N$ theory, Eq. (6.15) allows one to construct a relation between correlation functions of $n_\theta$ and $\sum_I g_{IIK}f(\theta)\phi_K$ at $\boldsymbol{q} = 0$. However, because $\phi_K$ has $N < N^2 = \dim \mathrm{U}(N)$ independent components, the "non-Abelian" and "trace" fluctuations of $g_{IJK}f(\theta)\phi_K$ are not independent, meaning that Eq. (6.15) *cannot* be used to fix the propagator of $g_{IJK}f(\theta)\phi_K$ and, furthermore, the optical conductivity exactly. This means that the multi-critical point described by this theory can subvert our earlier arguments that the anomaly fixes the $\boldsymbol{q} = 0$ current-current correlator at the IR fixed point to be independent of frequency. In our companion work, Ref. [63], we show that this is indeed the case for "loop current" order parameters.

# 7 Discussion

Theories of metallic quantum criticality have long been challenging to tame. In this work, we focused on what is likely the simplest class of such critical points; namely, those associated with the onset of a broken symmetry. We further specialized to the case where the ordered phase preserves the microscopic translation symmetry, so the corresponding order parameter lives at zero momentum. The critical theory for such a transition is described in terms of electronic modes near the Fermi surface coupled to long wavelength critical fluctuations of the bosonic order parameter. This theory has been the subject of a large number of studies over the years, and it is clear that critical order parameter fluctuations destroy the Landau quasiparticles while preserving the sharpness of the Fermi surface. Although controlled perturbative treatments of deformations to this theory have been developed, many physical aspects of these theories remain poorly understood.

The novelty of the present work was that, by exploiting the emergent symmetry and anomaly of a low energy model expected to capture the universal physics, we were able to make exact, non-perturbative statements about a number of physical properties. Examples included results for the optical conductivity, the zero-momentum boson propagator, and for the critical behavior of a number of thermodynamic susceptibilities upon approaching from the proximate Landau Fermi liquid metal.

Looking further afield, we anticipate that the general logic we used to extract universal non-perturbative conclusions about the low energy physics will be an important tool in the analysis of other, more complex, theories of metallic quantum criticality. It will be interesting, for instance, to examine the critical theories for density wave ordering in metals where the order parameter fluctuations couple most strongly to a set of specific points ('hot spots') on the Fermi surface. More challenging examples are situations in which the critical point is not simply described within the framework of a Fermi surface coupled to a gapless boson. Nevertheless, we expect the logic of this paper may shed light on all such theories.

## Acknowledgements

We thank Ehud Altman, Erez Berg, Andrey Chubukov, Luca Delacrétaz, Ilya Esterlis, Eduardo Fradkin, Haoyu Guo, Wei-Han Hsiao, Steve Kivelson, Avraham Klein, Sung-Sik Lee, Max Metlitski, Michael Mulligan, Aavishkar Patel, Sri Raghu, Subir Sachdev, Yoni Schattner, Boris Spivak, Xiaoyu Wang, and Cenke Xu for discussions. HG and DVE were supported by the Gordon and Betty Moore Foundation EPiQS Initiative through Grant

No. GBMF8684 at the Massachusetts Institute of Technology. DVE was also supported by the Gordon and Betty Moore Foundation EPiQS Initiative through Grant No. GBMF8683 at Harvard University. TS was supported by US Department of Energy grant DE- SC0008739, and partially through a Simons Investigator Award from the Simons Foundation. This work was also partly supported by the Simons Collaboration on Ultra-Quantum Matter, which is a grant from the Simons Foundation (651446, TS).

# A Regularization of the mid-IR theory

Although the ultimate choice of regularization has no impact on IR observables, it does affect the details of how they are calculated. In Section 3, we explicitly chose a regularization of the mid-IR theory, Eqs. (3.5) – (3.6), in which we set cutoffs for the fermion momentum perpendicular to the Fermi surface, $\Lambda_\perp$, and for frequency $\Lambda_\omega$, and we took $\Lambda_\perp \to \infty$ prior to $\Lambda_\omega \to \infty$. This choice respects gauge invariance of fields which couple to spatial $\mathrm{U}(1)_{\mathrm{patch}}$ currents, meaning, as shown in Section 3.6, that such fields are guaranteed to be massless without need of a finite bare mass, $m_c^2$. In perturbative calculations, this regularization corresponds to evaluating Feynman diagrams by integrating over momentum first, followed by frequency. This can be seen to result in a finite Drude weight but vanishing compressibility, and an additional background term, Eq. (3.46), must be added to ensure a finite compressibility (for further discussion of how the regularization choice enters diagrammatic calculations, see Ref. [30]). Such a dependence on the order of integration also arises in the 1D models discussed in Section 2.

Here we consider the opposite order of limits, where $\Lambda_\omega \to \infty$ before $\Lambda_\perp \to \infty$. This regularization can be associated with the physical microscopic model, Eq. (3.1), because the mid-IR effective theory can be obtained on expanding Eq. (3.1) in momenta close to the Fermi surface without any restriction on frequency. In this regularization, the boson must be tuned to criticality with a finite $m_c^2 \neq 0$, which will guarantee consistency with the results for the boson propagator and optical conductivity in Sections 3.4 and 3.5. Before proceeding, we comment that all of the conclusions below can be checked perturbatively by considering the one-loop contribution to the boson self-energy with frequency integrals performed before momentum integrals. Such a calculation matches our general anomaly considerations because ultimately the $\mathrm{U}(1)_{\mathrm{patch}}$ anomaly is, like other more familiar anomalies, one-loop exact.

To see the need for a finite bare mass term, one repeats the calculation of Section 3.6 up to Eq. 3.44. Because we are now taking $\Lambda_\omega \to \infty$ first while holding $\Lambda_\perp$ fixed, Eq. 3.44 can no longer be used, since we cannot shift fermion momenta above $\Lambda_\perp$. Instead, however, we

are now free to perform a frequency shift,

$$\psi_\theta \to \exp\left(i\mu_\theta\, t\right)\psi_\theta\,. \tag{A.1}$$

Importantly, this shift *does not* eliminate the probe, because it is anomalous. Under this change of variables, the path integral measure transforms such that action picks up a term,

$$\delta S = -\int_{t,\boldsymbol{x}}\sum_\theta \mu_\theta t\,\frac{\Lambda_{\mathrm{patch}}}{(2\pi)^2}\frac{g^a(\theta)}{v_F(\theta)}\partial_t\phi_a = \int_{t,\boldsymbol{x}}\sum_\theta \mu_\theta\,\frac{\Lambda_{\mathrm{patch}}}{(2\pi)^2}\frac{g^a(\theta)}{v_F(\theta)}\phi_a\,, \tag{A.2}$$

where the second equality follows from integration by parts. Consequently, we find that in this regularization,

$$\chi_{\psi_\theta^\dagger\psi_\theta,\phi_b} = \sum_a \frac{\Lambda_{\mathrm{patch}}}{(2\pi)^2}\frac{g^a(\theta)}{v_F(\theta)}\chi_{\phi_a\phi_b}\,, \tag{A.3}$$

where we have made the sum over $a$ explicit for clarity. Plugging this into Eq. (3.42) (noting that in the current regularization, we should replace $n_\theta$ in that equation with $\psi_\theta^\dagger\psi_\theta$) and promote $m_c^2$ to a matrix in boson flavor space, denoted $R^{ab}$, we obtain

$$\sum_c R^{ac}\chi_{\phi_c\phi_b} = \delta^{ab} + \sum_c\sum_\theta \frac{\Lambda_{\mathrm{patch}}}{(2\pi)^2}\frac{g^a(\theta)g^c(\theta)}{v_F(\theta)}\chi_{\phi_c\phi_b} \tag{A.4}$$

$$= \delta^{ab} + \sum_c \Pi^{ac}\chi_{\phi_c\phi_b}\,, \tag{A.5}$$

where $\Pi^{ab}$ is as defined in Eq. (3.18). We therefore find

$$\chi_{\phi_a\phi_b} = [R-\Pi]^{-1}_{ab}\,. \tag{A.6}$$

As a result, criticality is achieved on tuning $R^{ab} = \Pi^{ab}$, which is finite. Notice that for the case where $\phi$ is the spatial component of a U(1) gauge field, where $g^a(\theta)$ replaced with $v_F(\theta)w^i(\theta)$ and $\phi_i$ now has a spatial index, the term $\frac{1}{2}\Pi^{ij}\phi_i\phi_j$ is none other than the mid-IR realization of the diamagnetic term appearing in the UV theory in Eq. 3.1. However, we emphasize that, in this order of limits, the mid-IR theory at criticality should contain a bare boson mass regardless of whether the microscopic theory contains a diamagnetic term.

The need for a finite mass for $\phi_a$ at criticality means that, rather than coupling to $g^a(\theta)\psi_\theta^\dagger\psi_\theta$ alone, $\phi_a$ couples at criticality as

$$S_{\phi\psi} = \int_{t,\boldsymbol{x}}\phi_a\left(g^a(\theta)\,\psi_\theta^\dagger\psi_\theta - \frac{1}{2}\Pi^{ab}\phi_b\right)\,. \tag{A.7}$$

Now, say that we continue to view $\phi_a$ as a vector potential for some subgroup of $\mathrm{U}(1)_{\text{patch}}$, as in the main text (we describe the meaning of this statement more precisely in Appendix B). This indicates that the "gauge invariant" chiral density operator that $g^a(\theta)\phi_a$ couples to is actually

$$n_\theta = \psi_\theta^\dagger \psi_\theta - \frac{\Lambda_{\text{patch}}}{(2\pi)^2} \frac{g^a(\theta)}{v_F(\theta)} \phi_a\,. \tag{A.8}$$

We term this operator as gauge invariant because the second term is required to ensure that the low energy theory possesses a gauge invariance for $\phi^a$ to compensate for the fact that taking $\Lambda_\perp$ to be finite explicitly breaks the gauge symmetry. This suggests that we should identify the $n_\theta$ operator in the anomaly equation, Eq. (3.12), with the operator in Eq. (A.8). Indeed, we will see below that this identification is consistent with the discussion above and leads to $n_\theta$ having the same susceptibilities in the present regularization as the $n_\theta$ operator discussed in Section 3.6, which used the opposite order of limits where $\Lambda_\perp \to \infty$ first.

Because we define $\phi$ to couple as the spatial component of a $\mathrm{U}(1)_{\text{patch}}$ gauge field and $\mu_\theta$ to couple as the *temporal* component, at $\omega = 0$ one finds a corrected anomaly equation,

$$\boldsymbol{w}(\theta) \cdot \boldsymbol{\nabla}(v_F(\theta)\, n_\theta) = \frac{\Lambda_{\text{patch}}}{(2\pi)^2}\, \boldsymbol{w}(\theta) \cdot \boldsymbol{\nabla}\mu_\theta\,, \tag{A.9}$$

where we have redefined the density probe, $\mu_\theta(\boldsymbol{w} \cdot \boldsymbol{x})$, such that it couples as $\mu_\theta\, n_\theta$ (rather than to $\psi_\theta^\dagger \psi_\theta$ alone) and weakly depends on the spatial direction perpendicular to the Fermi surface. Therefore, the right hand side of Eq. (A.9) can be understood as the electric field due to $\mu_\theta$, and the left hand side is $\boldsymbol{\nabla} \cdot \boldsymbol{j}_\theta$. Fourier transforming to $q_\perp(\theta) = \boldsymbol{w}(\theta) \cdot \boldsymbol{q}$ and dividing out by momentum, we find the relation at $\omega = 0$,

$$\begin{aligned}
\langle n_\theta \rangle_{\mu_\theta} &= \left\langle \psi_\theta^\dagger \psi_\theta(\omega = 0, q_\perp) - \frac{\Lambda_{\text{patch}}}{(2\pi)^2} \frac{g^a(\theta)}{v_F(\theta)} \phi_a(\omega = 0, q_\perp) \right\rangle_{\mu_\theta} \\
&= \frac{\Lambda_{\text{patch}}}{(2\pi)^2} \frac{1}{v_F(\theta)} \mu_\theta(\omega = 0, q_\perp)\,.
\end{aligned} \tag{A.10}$$

Importantly, the second line indicates that the relation between susceptibilities of $\psi_\theta^\dagger \psi_\theta$ and $\phi_a$, which we obtained using the change of variables in Eq. (A.1), misses the effect of a term depending on probes. Such a term should be included in computing static susceptibilities of $n_\theta$, for which we take $\omega \to 0$ followed by $\boldsymbol{q} \to 0$.

We can use Eq. (A.10) to immediately calculate the susceptibility, $\chi_{n_\theta n_{\theta'}}$,

$$\chi_{n_\theta n_{\theta'}} = \frac{\delta \langle n_\theta \rangle_{\mu_{\theta'}}}{\delta \mu_{\theta'}} = \frac{\Lambda_{\text{patch}}}{(2\pi)^2\, v_F(\theta)} \delta_{\theta,\theta'}\,, \tag{A.11}$$

which indeed matches Eq. (3.48). The susceptibility for the conserved charge density, $\chi_{\tilde{n}_\theta \tilde{n}_\theta}$, can be obtained using the definition in Eq. (3.13). If we consider deviations from criticality of the form $m_c^2 \, \delta^{ab} = R^{ab} - \Pi^{ab}$, we obtain

$$\chi_{\tilde{n}_\theta \tilde{n}_{\theta'}} = \frac{\Lambda_{\text{patch}}(\theta)}{(2\pi)^2 v_F(\theta)} \, \delta_{\theta\theta'} + \frac{\Lambda_{\text{patch}}(\theta) \, \Lambda_{\text{patch}}(\theta')}{(2\pi)^4} \frac{g^a(\theta) g_a(\theta')}{v_F(\theta) v_F(\theta')} \frac{1}{m_c^2} \,, \tag{A.12}$$

which matches Eq. (3.50).

Finally, we briefly remark on how the details of the derivation of Section 3.4 change in the present regularization. As asserted above, $n_\theta$ still satisfies the same anomaly equation, Eq. (3.14). However, because of the different relationship between $n_\theta$ and $\psi_\theta^\dagger \psi_\theta$, the equation of motion Eq. (3.16) takes a different form when expressed in terms of $n_\theta$. Consequently, in contrast to the result of Section 3.4, one would conclude that the boson self-energy at $\boldsymbol{q} = 0$ *vanishes* (this is consistent with what one finds computing the one-loop self energy diagram integrating over frequency before momentum). Thus, in terms of the bare boson mass matrix $R$, one obtains

$$D_{ab}(\omega) = [-\lambda \omega^2 \, \mathbb{I} + R]^{-1}{}_{ab}. \tag{A.13}$$

However, if we use the fact shown above that in the regularization of the current appendix, $R = \Pi$ at criticality, whereas in the result Eq. (3.19) obtained in the other regularization, we should set $m_c^2 = 0$ at criticality, we see that the results Eq. (A.13) and Eq. (3.19) are ultimately identical upon tuning to the critical point. Similar results hold for the optical conductivity derived in Section 3.5.

Thus, we have seen that the regularization in which $\Lambda_\omega \to \infty$ before $\Lambda_\perp \to \infty$ realizes all of the same physics derived in the main text in the opposite order or limits, but with a finite bare boson mass at criticality.

# B   Path integral derivation of the U(1)$_{\text{patch}}$ anomaly

In this Appendix, we derive the $U(1)_{\text{patch}}$ anomaly in Eq. (3.12) by considering the transformation properties of the path integral measure in the Euclidean partition function,

$$Z = \int \mathcal{D}\phi_a \, e^{-S_{\text{boson}}} \prod_\theta Z_{\text{patch}}[\phi_a; \theta] \,, \ Z_{\text{patch}}[\phi_a; \theta] = \int \mathcal{D}\psi_\theta^\dagger \mathcal{D}\psi_\theta \, e^{-S_{\text{patch}}(\theta)} \,, \tag{B.1}$$

where $S_{\text{patch}}$ is the Euclidean version of Eq. (3.6). Our analysis follows the standard approach originally developed by Fujikawa to derive the axial anomaly [69] (for a review, see Ref. [56]). Note that for most of this Appendix we will Wick rotate to imaginary time, $\tau$, where $t = -i\tau$.

Our focus will be on the partition function for a single patch, $Z_{\text{patch}}[\phi_a; \theta]$. Consider a local (in spacetime) $U(1)$ rotation of each $\psi_\theta$,

$$\psi_\theta \to \psi'_\theta = e^{iq_\theta \alpha(\tau, \boldsymbol{x})} \psi_\theta \,. \tag{B.2}$$

Here $q_\theta$ defines a particular sub-group of $U(1)_{\text{patch}}$, e.g. $q_\theta = 1$ for all $\theta$ corresponds to the physical EM charge conservation symmetry. In particular, our interest is in a class of "axial" rotations that satisfy,

$$\sum_\theta q_\theta = 0 \,. \tag{B.3}$$

For a model with only two patches, such transformations are the usual axial rotations. Under this transformation, the action, $S_{\text{patch}}$ transforms classically as

$$\delta_{(0)} S_{\text{patch}} \tag{B.4}$$

$$= i \int_{\tau,\boldsymbol{x}} \sum_\theta q_\theta \, \alpha(\tau, \boldsymbol{x}) \left\{ \partial_\tau (\psi'^\dagger_\theta \psi'_\theta) - i\partial_i [v_F(\theta) w^i(\theta) \psi'^\dagger_\theta \psi'_\theta] - \frac{\kappa^{ij}(\theta)}{2} \partial_i (\psi'^\dagger_\theta \partial_j \psi'_\theta + \text{h.c.}) \right\} \,. \tag{B.5}$$

In the quantum theory, there is also a contribution from the Jacobian of the path integral. For each patch, we expand the fermion fields in an arbitrary normalizeable basis $\varphi_{\theta,n}$

$$\psi_\theta = \sum_n a_n(\theta) \, \varphi_{\theta,n} \,, \ \psi^\dagger_\theta = \sum_n \overline{a}_n(\theta) \, \varphi^\dagger_{\theta,n} \,. \tag{B.6}$$

Here $\varphi_{\theta,n}$ form a complete orthonormal set of c-number valued functions, and $a_n(\theta)$ are Grassmann numbers (the index $n$ may be discrete or continuous).

Now we may write the fermionic part of the path integral measure as

$$\prod_\theta \mathcal{D}\psi^\dagger_\theta \, \mathcal{D}\psi_\theta = \prod_\theta \prod_n d\overline{a}_n(\theta) \, da_n(\theta) \,. \tag{B.7}$$

Under the change of variables in Eq. (B.2), the coefficients, $a_n(\theta)$, transform to new coefficients, $a'_n(\theta)$, given by

$$a'_n(\theta) = \int_{\boldsymbol{x},\tau} \varphi^\dagger_{\theta,n} \sum_m (1 + iq_\theta \alpha) a_m(\theta) \varphi_{\theta,m} + \mathcal{O}(\alpha^2) \,. \tag{B.8}$$

We drop the terms that are $\mathcal{O}(\alpha^2)$ below. Changing variables to $a'_n(\theta)$ in the path integral thus introduces a Jacobian,

$$\prod_\theta \mathcal{D}\psi^\dagger_\theta \, \mathcal{D}\psi_\theta = \prod_\theta \prod_n d\overline{a}'_n(\theta) \, da'_n(\theta) \left[ \det\left(\mathbb{I} + C\right) \right]^2 \,, \ C^{\theta\theta'}_{nm} = \delta^{\theta\theta'} \int_{\boldsymbol{x},\tau} \varphi^\dagger_{\theta,n} \, iq_\theta \alpha \, \varphi_{\theta,m} \,, \tag{B.9}$$

where we do not sum over $\theta$ in the definition of $C$. Our goal is to evaluate the Jacobian. We first use the standard identity, $\log \det = \operatorname{Tr} \log$, to write

$$[\det (\mathbb{I} + C)]^2 = \exp\left[2 \operatorname{Tr} \log (\mathbb{I} + C)\right] = \exp\left[2 \operatorname{Tr} C + \mathcal{O}(\alpha^2)\right] \tag{B.10}$$

$$= \exp\left[2i \int_{\boldsymbol{x},\tau} \alpha(\tau, \boldsymbol{x}) \sum_{n,\theta} \varphi_{n,\theta}^\dagger \, q_\theta \, \varphi_{n,\theta}\right] . \tag{B.11}$$

The sum in the exponential is a trace over a complete set of functions. For convenience of computations later, we will choose $\varphi_{\theta,n}$ to be a basis of plane waves so that

$$\sum_{n,\theta} \varphi_{n,\theta}^\dagger \, q_\theta \, \varphi_{n,\theta} = \sum_\theta \int_{\boldsymbol{k},\omega} q_\theta . \tag{B.12}$$

Evaluating the above expression as-is is problematic: the sum over $\theta$ yields zero, yet the integral over momentum and frequency gives infinity. We therefore need to regulate this expression.

As in Fujikawa's original derivation for Dirac fermions, we choose a Gaussian regularization, which we refer to below as a "soft cutoff," in which we suppress the sum in Eq. (B.12) at large eigenvalues of a suitable energy operator which we will define below. This regularization amounts to only choosing soft cutoffs for the frequency ($\Lambda_\omega$) and the momentum perpendicular to the Fermi surface ($\Lambda_\perp$). In particular, we will choose $\Lambda_\omega = v_F(\theta)\Lambda_\perp \equiv \Lambda \ll k_F$. For the momentum tangent to the Fermi surface, we choose a *hard cutoff*, $\Lambda_{\text{patch}}$, which constitutes the width of the patch[14]. For simplicity, we will assume that this cutoff is the same on each patch, although in principle the cutoff on each patch can be treated independently. At the end of the calculation, we will take the limit

$$\Lambda \to \infty \,, \quad \frac{\Lambda_{\text{patch}}}{\Lambda} \text{ fixed.} \tag{B.13}$$

In this regularization, Eq. (B.12) is replaced with

$$\sum_{n,\theta} \varphi_{n,\theta}^\dagger \, q_\theta \, \varphi_{n,\theta} \to \sum_\theta \int_{\Lambda_{\text{patch}}} \frac{dk_\parallel(\theta)}{2\pi} \int \frac{dk_\perp(\theta)d\omega}{(2\pi)^2} \frac{1}{v_F(\theta)} \, q_\theta \, e^{i\omega\tau - i\boldsymbol{k}\cdot\boldsymbol{x}} \, e^{-\mathcal{H}_\theta/\Lambda^2} \, e^{-i\omega\tau + i\boldsymbol{k}\cdot\boldsymbol{x}} ,$$

$$\tag{B.14}$$

---

[14]This choice of regulator explicitly breaks gauge invariance of fields coupling to the component of the current in the direction tangent to the Fermi surface. However, in principle, one could attempt to construct a different, "soft" analogue of the $\Lambda_{\text{patch}}$ regulator. Nevertheless, since our focus in this work is on fields coupling to the perpendicular component of the current, this will not affect the general result for the anomaly.

where $\omega$ is the Matsubara frequency; $k_\parallel(\theta) = \varepsilon_{ij}w^i(\theta)k^j$ and $k_\perp(\theta) = v_F(\theta)w^i(\theta)k_i$ are respectively plane wave momenta tangent and perpendicular to the Fermi surface at patch $\theta$. The operator, $\mathcal{H}_\theta$, defining the regulator is the square of analogues of Euclidean "Dirac operators," which we will define below.

Importantly, our choice of regularization will be such that it preserves invariance under gauge transformations of the type,

$$\psi_\theta \to e^{if(w^i(\theta)x_i)}\psi_\theta \, , \, g^a(\theta)\phi_a \to g^a(\theta)\phi_a + v_F(\theta)w^i(\theta) \, \partial_i f \, . \tag{B.15}$$

We emphasize that such transformations may correspond to a different subgroup of $U(1)_{\text{patch}}$ from the physical EM symmetry, which we also require invariance under (we implicitly couple to a background EM gauge field as well). We will see at the end of the calculation that this is possible if the theory satisfies the condition,

$$\sum_\theta \frac{g^a(\theta)}{v_F(\theta)} = 0 \, . \tag{B.16}$$

One recognizes this as the condition that $\sum_\theta n_\theta = \sum_\theta \widetilde{n}_\theta$, where $\widetilde{n}_\theta$ is the conserved charge density defined in Eq. (3.13), correspond to the same physical charge density. We remark that this condition is actually enforced by symmetry in the case of a system with inversion symmetry where the order parameter is inversion-odd, or in the case of a system with $C_4$ symmetry and an Ising-nematic order parameter, provided that the patches are chosen compatible with the symmetry. For example, for a loop current order parameter, one only needs an even number of patches to ensure this because $g^a(\theta) = -g^a(\theta + \pi)$. In contrast, for an Ising-nematic order parameter with $C_4$ symmetry, the number of patches must be a multiple of four since each pair of antipodal patches has the same value of $g^a(\theta) = g^a(\theta + \pi)$. Note that these statements assume each patch has equal width.

We now proceed to define the operator, $\mathcal{H}_\theta$, appearing in the Gaussian regulator in Eq. (B.14). Let each index, $\theta \in [0, \pi)$, label the pair of antipodal patches $\theta$, $\theta + \pi$. We may construct Euclidean gamma matrices acting on the antipodal patch indices as follows,

$$\gamma^0 = \sigma^y \, , \, \gamma^1 = \sigma^x \, , \, \gamma_\star = i\gamma^0\gamma^1 = \sigma^z. \tag{B.17}$$

Using $\partial_\perp(\theta)$ as a shorthand for the perpendicular derivative $v_F(\theta)w^i(\theta)\partial_i$, the Euclidean Dirac operator $\mathcal{D}(\theta)$ for the two patches $\theta$, $\theta + \pi$, can then be defined as

$$\mathcal{D}(\theta) = i\gamma^\mu D_\mu(\theta) \, , \tag{B.18}$$

where the matrices $D_0(\theta), D_1(\theta)$ are

$$D_0(\theta) = \mathbb{I}\,\partial_\tau\,,\ D_1(\theta) = \begin{pmatrix} -\partial_\perp(\theta) + ig^a(\theta)\phi_a & 0 \\ 0 & \partial_\perp(\theta+\pi) - ig^a(\theta+\pi)\phi_a \end{pmatrix}, \tag{B.19}$$

with $\mathbb{I}$ the identity matrix in the two-patch space. Note that we have defined this operator such that $g^a(\theta)\phi_a$ couples like a *vector* potential. We emphasize that for order parameters which are even under inversions, $g^a(\theta) = g^a(\theta+\pi)$ (such as Ising-nematic), this choice of covariant derivative would lead to an inconsistency unless the total number of patches is a multiple of four (for the case of $C_4$ symmetry), as discussed above around Eq. (B.16). This can be most straightforwardly seen in an example with only two patches, $\theta = 0, \pi$. In that case, such an order parameter would couple to $\sum_\theta n_\theta = \psi_0^\dagger \psi_0 + \psi_\pi^\dagger \psi_\pi$, which we choose to identify as the physical charge density, but is the axial current. If Eq. (B.19) were used in this example, one would ultimately find that the physical EM symmetry is anomalous and $\sum_\theta n_\theta$ is not a conserved charge density, although $\sum_\theta \widetilde{n}_\theta$ of Eq. (3.13) would still be conserved. Therefore, unless one works with a multiple of four patches, consistency with conservation of $\sum_\theta n_\theta$ requires including inversion-even order parameters in the *temporal* component of the covariant derivative, thus making them *scalar* potentials. Throughout the main text, our assumption was that the total number of patches is always chosen such that Eq. (B.19) is a consistent choice.

Because for antipodal pairs, $\mathcal{D}(\theta)$ is simply the usual 1+1-D Dirac operator, its square is diagonal, with entries

$$[\mathcal{D}(\theta)]^2 = -\begin{pmatrix} \partial_\tau^2 + \det D_1(\theta) + g^a(\theta)\,\partial_\tau\phi_a & 0 \\ 0 & \partial_\tau^2 + \det D_1(\theta) + g^a(\theta+\pi)\,\partial_\tau\phi_a \end{pmatrix}. \tag{B.20}$$

The operator appearing in the regulated sum of Eq. (B.14), $\mathcal{H}_\theta$, can finally be constructed as

$$\mathcal{H}_\theta = -\begin{cases} \partial_\tau^2 + \det D_1(\theta) + g^a(\theta)\,\partial_\tau\phi_a & \theta \in [0, \pi) \\ \partial_\tau^2 + \det D_1(\theta-\pi) + g^a(\theta)\,\partial_\tau\phi_a & \theta \in [\pi, 2\pi) \end{cases}. \tag{B.21}$$

Plugging the regulator Eq. (B.21) into Eq. (B.14) and expanding in powers of $\phi$, one finds a leading divergent term proportional to

$$\sum_\theta \frac{1}{v_F(\theta)}\, q_\theta\, \Lambda_{\text{patch}}\, \Lambda^2\,, \tag{B.22}$$

which vanishes by the requirement, $\sum_\theta q_\theta = 0$. The next leading, nonvanishing term is linear in $\phi$

$$\sum_{n,\theta} q_\theta\, \varphi^\dagger_{n,\theta}\varphi_{n,\theta}(\tau,\boldsymbol{x}) = \sum_\theta q_\theta \int_{\Lambda_{\text{patch}}} \frac{dk_\|(\theta)}{2\pi} \int_{k_\perp(\theta),\omega} \frac{1}{\Lambda^2}\frac{g^a(\theta)\,\partial_\tau\phi_a}{v_F(\theta)}\, e^{-(\omega^2 + k_\perp^2(\theta))/\Lambda^2} \quad \text{(B.23)}$$

$$= -i\,\frac{\Lambda_{\text{patch}}}{2(2\pi)^2}\sum_\theta q_\theta\,\frac{g^a(\theta)}{v_F(\theta)}\,\partial_t\phi_a + \mathcal{O}(\Lambda_{\text{patch}}/\Lambda^2)\,, \quad \text{(B.24)}$$

where we have Wick rotated back to real time using $t = -i\tau$. Note that there is also a term proportional to $\sum_\theta q_\theta v_F(\theta) w^i(\theta) g^a(\theta)\partial_i\phi_a$ that comes from expanding $\det D_1(\theta)$. This term would be nonvanishing if $\phi$ instead coupled like a *scalar* potential. To guarantee consistency of the aforementioned requirement that $\phi_a$ instead couples like a vector potential, we therefore need to impose a final requirement,

$$\sum_\theta q_\theta\, w^i(\theta) g^a(\theta) = 0\,. \quad \text{(B.25)}$$

We will return to the implications of imposing this requirement below.

All further terms vanish in the limit $\Lambda \to \infty$. Therefore, under the change of variables in Eq. (B.2), the (real time) action changes by

$$\delta S = \int_{\boldsymbol{x},t} \alpha(t,\boldsymbol{x}) \sum_\theta q_\theta \left( \partial_\mu j^\mu_\theta + \frac{\Lambda_{\text{patch}}}{(2\pi)^2}\frac{g^a(\theta)}{v_F(\theta)}\,\partial_t\phi_a \right)\,. \quad \text{(B.26)}$$

Here $j^\mu_\theta = (n_\theta = \psi^\dagger_\theta\psi_\theta, j^i_\theta)$. The anomalous Ward identities are derived by differentiating with respect to $\alpha$. In particular, $\sum_\theta q_\theta j_\theta$ is not conserved. Instead,

$$\sum_\theta \partial_\mu(q_\theta j^\mu_\theta) = -\frac{\Lambda_{\text{patch}}}{(2\pi)^2}\sum_\theta q_\theta\,\frac{g^a(\theta)}{v_F(\theta)}\,\partial_t\phi_a\,. \quad \text{(B.27)}$$

This is the analogue of the axial anomaly for the "mid-IR" theory discussed in this work. If Eq. (B.27) holds with $q_\theta$ chosen to be arbitrary, then we would obtain the anomaly equation from the main text:

$$\partial_\mu j^\mu_\theta = -\frac{\Lambda_{\text{patch}}}{(2\pi)^2}\frac{g^a(\theta)}{v_F(\theta)}\,\partial_t\phi_a\,. \quad \text{(B.28)}$$

However, some caution is needed since $q_\theta$ above was *not* arbitrary but rather was required to satisfy the conditions Eq. (B.3) and Eq. (B.25). These requirements were imposed such that (1) the $q_\theta$'s correspond to an anomalous subgroup of $U(1)_{\text{patch}}$ (one may think of them as the eigenvalues of a generalization of $\gamma_\star$ to the mid-IR theory), and (2) the theory respects

gauge invariance under Eq. (B.15) (which is *not* anomalously broken). This is a limitation of the Fujikawa procedure also present in the usual 1D example, and we nevertheless expect that the anomaly equation, Eq. (B.28), will hold generally.

In any case, the more limited Eq. (B.27) would be sufficient for many arguments in this paper. In particular, for the boson propagator derivation, it is sufficient that Eq. (B.27) holds with $q_\theta = g^b(\theta)$ (for arbitrary index, $b$), while for the optical conductivity derivation, we additionally require that Eq. (B.27) holds with $q_\theta = v_F(\theta)w^i(\theta)$ (for arbitrary index $i$). If we additionally invoke the fact that the total charge is always conserved, then these requirements can be seen to hold both in the case of an inversion-symmetric system where the order parameter is odd under inversion, and in the case of a system with a $C_4$ rotation system with an Ising-nematic order parameter. These were the same symmetries that imposed Eq. (B.16).

Note that the regularization used here to compute the Jacobian of the path integral is somewhat different from the one discussed in the previous Appendix and in the main text, where different cutoffs were implemented for $k_\perp(\theta)$ and $\omega$, and we chose an order of limits in which $\Lambda_\perp \to \infty$ first. The ultimate conclusions in both here and with the regularization used in the main text should be the same (in the absence of scalar potentials), as both the Gaussian regularization used here and the $\Lambda_\perp \to \infty$ first regularization preserve gauge invariance if $\phi_a$ is treated like a *vector* potential.

## C  Fundamental large $N$ calculation of the boson self energy

In this appendix, we provide a detailed computation of the boson self energy $\Pi(\boldsymbol{q} = 0, i\nu)$ within the RPA expansion. As explained in Section 5.2, to understand the essence of the forthcoming diagrammatic calculations, it is sufficient to consider a mid-IR action with scalar $\phi, g(\theta)$

$$S = S_{\text{boson}} + \sum_\theta S_{\text{patch}}(\theta) \,, \tag{C.1}$$

where

$$S_{\text{patch}}(\theta) = \int_{\tau,\boldsymbol{x}} \left[ \psi_\theta^\dagger \left\{ \partial_\tau - v_F(\theta)\boldsymbol{w}(\theta) \cdot \boldsymbol{k} + \kappa_{ij}(\theta)k_i k_j \right\} \psi_\theta + g(\theta)\phi \, \psi_\theta^\dagger \psi_\theta \right] \,, \tag{C.2}$$

$$S_{\text{boson}} = \frac{N_f}{2} \int \phi(\boldsymbol{q}, \tau)[-\lambda \partial_\tau^2 + |\boldsymbol{q}|^{z-1}]\phi(\boldsymbol{q}, \tau) \,. \tag{C.3}$$

We now make a brief comment on regularizations. For the theory of a full Fermi surface, there is no need to regularize the fermion momentum integrals, as the dominant contributions

come from a small neighborhood around a compact Fermi surface. However, when we go into a patch $\theta$ and choose coordinates $k_\perp/k_\parallel$ that are perpendicular/parallel to the patch, we must impose a hard cutoff $-\Lambda_\parallel < k_\parallel < \Lambda_\parallel$. In order to satisfy this requirement for all scattering processes, the interaction term must be carefully written as

$$\int^{\Lambda_\omega} d\omega d\Omega \int^{\Lambda_\perp} dq_\perp dk_\perp \int_{-2\Lambda_\parallel}^{2\Lambda_\parallel} dq_\parallel \int_{-\Lambda_\parallel + \frac{|q_\parallel|}{2}}^{\Lambda_\parallel - \frac{|q_\parallel|}{2}} dk_\parallel \phi(\boldsymbol{q},\Omega)\psi^\dagger(\boldsymbol{k}+\frac{\boldsymbol{q}}{2},\omega+\frac{\Omega}{2})\psi(\boldsymbol{k}-\frac{\boldsymbol{q}}{2},\omega-\frac{\Omega}{2}).$$
(C.4)

In all loop integrals, we will send the frequency cutoff $\Lambda_\omega \to \infty$ before sending the $k_\perp$ cutoff $\Lambda_\perp \to \infty$, while maintaining a finite $\Lambda_\parallel$ (this is a shorthand for the cutoff $\Lambda_{\text{patch}}$ defined in the main text). This regularization scheme is invariant under $\boldsymbol{k} \to -\boldsymbol{k}$ and $\boldsymbol{q} \to -\boldsymbol{q}$, which guarantees the Hermiticity of the Hamiltonian associated with Eq. (C.4).

## C.1 Diagrammatic rules in the mid-IR theory

In the RPA limit, the fermion propagator within each patch remains free while the boson propagator takes a Landau damping form

$$G_\theta(\boldsymbol{k},i\omega) = \frac{1}{i\omega - \epsilon_\theta(\boldsymbol{k})} \quad D(\boldsymbol{q},i\nu) = \frac{1}{|\boldsymbol{q}|^{z-1} + \gamma_{\hat{q}}\frac{|\nu|}{|\boldsymbol{q}|}},$$
(C.5)

where $\gamma_{\hat{q}}$ generally depends on the orientation $\hat{q}$ and $\epsilon_\theta(\boldsymbol{k}) = v_F(\theta)\boldsymbol{w}(\theta)\cdot\boldsymbol{k} + \kappa_{ij}(\theta)k_ik_j$ is the fermion dispersion in patch $\theta$. Using $G_\theta, D$ in internal loops, we can then compute $\Pi(q=0,i\nu)$ in a diagrammatic expansion in powers of $1/N_f$.

To leading order, there is a single one-loop free fermion bubble $\Pi^{(0)}$ which gives the expected anomaly prediction $\frac{\Lambda_\parallel}{(2\pi)^2}\sum_\theta \frac{g(\theta)^2}{v_F(\theta)}$ (this is a standard calculation which we will not repeat). The leading $\mathcal{O}(1/N_f)$ corrections come from five diagrams: the self energy diagrams $\Pi^{(\text{SE},1)}/\Pi^{(\text{SE},2)}$, the vertex correction (Maki-Thompson) diagram $\Pi^{(\text{MT})}$, and the Aslamazov-Larkin diagrams $\Pi^{(\text{AL},1)}/\Pi^{(\text{AL},2)}$ (see Figure 4). In what follows, we will show that

$$\Pi^{(\text{SE},1)} + \Pi^{(\text{SE},2)} + \Pi^{(\text{MT})} = \Pi^{(\text{AL},1)} + \Pi^{(\text{AL},2)} = 0,$$
(C.6)

for both Ising-nematic and loop current order parameters, thereby demonstrating the consistency of RPA with the anomaly to this order. Throughout the calculation, two identities will be used repeatedly and we collect them here for convenience:

$$\frac{G_\theta\left(\boldsymbol{k}-\frac{\boldsymbol{q}}{2},i\omega\right) - G_\theta\left(\boldsymbol{k}+\frac{\boldsymbol{q}}{2},i\omega+i\nu\right)}{\left[i\nu - \epsilon_\theta\left(\boldsymbol{k}+\frac{\boldsymbol{q}}{2}\right) + \epsilon_\theta\left(\boldsymbol{k}-\frac{\boldsymbol{q}}{2}\right)\right]} = G_\theta\left(\boldsymbol{k}-\frac{\boldsymbol{q}}{2},i\omega\right)G_\theta\left(\boldsymbol{k}+\frac{\boldsymbol{q}}{2},i\omega+i\nu\right),$$
(C.7)

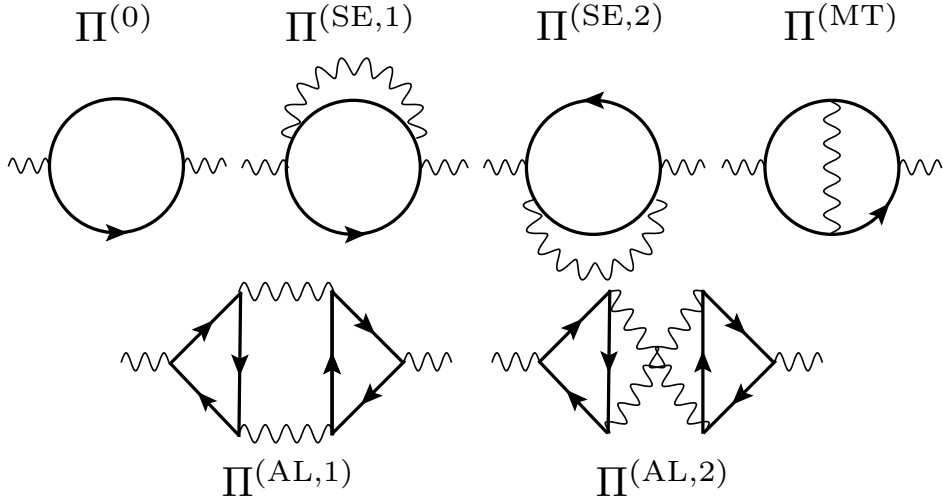

**Figure 4:** Diagrammatic contributions to the boson self energy, $\Pi(\boldsymbol{q} = 0, i\nu)$, to $\mathcal{O}(1/N_f)$. $\Pi^{(0)}$ is the one-loop RPA bubble. $\Pi^{(\mathrm{SE},1)}, \Pi^{(\mathrm{SE},2)}$, and $\Pi^{(\mathrm{MT})}$ are the fermion self energy corrections and the vertex correction (Maki-Thompson diagram). $\Pi^{(\mathrm{AL},1)}, \Pi^{(\mathrm{AL},2)}$ are the Aslamasov-Larkin diagrams.

$$\int \frac{d\omega}{2\pi} G_\theta \left( \boldsymbol{k} - \frac{\boldsymbol{q}}{2}, i\omega \right) - G_\theta \left( \boldsymbol{k} + \frac{\boldsymbol{q}}{2}, i\omega + i\nu \right) = \frac{1}{2} \left( \mathrm{sgn}\left[ \epsilon_\theta \left( \boldsymbol{k} + \frac{\boldsymbol{q}}{2} \right) \right] - \mathrm{sgn}\left[ \epsilon_\theta \left( \boldsymbol{k} - \frac{\boldsymbol{q}}{2} \right) \right] \right) .$$
(C.8)

## C.2   Cancellation between self energy and vertex diagrams

The contributions from $\Pi^{(\mathrm{SE},1)}, \Pi^{(\mathrm{SE},2)}, \Pi^{(\mathrm{MT})}$ each contains a single fermion loop and comes with a $-1$ prefactor. Using the general identity (C.7), we can easily show that

$$
\begin{aligned}
\Pi^{(\mathrm{SE},1)} &= -g(\theta)^4 \int G_\theta \left( \boldsymbol{k} - \frac{\boldsymbol{q}'}{2}, i\omega \right)^2 G_\theta \left( \boldsymbol{k} - \frac{\boldsymbol{q}'}{2}, i\omega + i\nu \right) G_\theta \left( \boldsymbol{k} + \frac{\boldsymbol{q}'}{2}, i\omega + i\nu' \right) D(\boldsymbol{q}', i\nu') \\
&= -\frac{g(\theta)^4}{i\nu} \int G_\theta \left( \boldsymbol{k} - \frac{\boldsymbol{q}'}{2}, i\omega \right) G_\theta \left( \boldsymbol{k} + \frac{\boldsymbol{q}'}{2}, i\omega + i\nu' \right) \\
&\qquad\qquad \cdot \left[ G_\theta \left( \boldsymbol{k} - \frac{\boldsymbol{q}'}{2}, i\omega \right) - G_\theta \left( \boldsymbol{k} - \frac{\boldsymbol{q}'}{2}, i\omega + i\nu \right) \right] D(\boldsymbol{q}', i\nu') ,
\end{aligned}
$$
(C.9)

$$
\begin{aligned}
\Pi^{(\mathrm{SE},2)} &= -g(\theta)^4 \int G_\theta \left( \boldsymbol{k} - \frac{\boldsymbol{q}'}{2}, i\omega - i\nu \right) G_\theta \left( \boldsymbol{k} - \frac{\boldsymbol{q}'}{2}, i\omega \right)^2 G_\theta \left( \boldsymbol{k} + \frac{\boldsymbol{q}'}{2}, i\omega + i\nu' \right) D(\boldsymbol{q}', i\nu') \\
&= -\frac{g(\theta)^4}{i\nu} \int G_\theta \left( \boldsymbol{k} - \frac{\boldsymbol{q}'}{2}, i\omega + i\nu \right) G_\theta \left( \boldsymbol{k} + \frac{\boldsymbol{q}'}{2}, i\omega + i\nu + i\nu' \right) \\
&\qquad\qquad \cdot \left[ G_\theta \left( \boldsymbol{k} - \frac{\boldsymbol{q}'}{2}, i\omega \right) - G_\theta \left( \boldsymbol{k} - \frac{\boldsymbol{q}'}{2}, i\omega + i\nu \right) \right] D(\boldsymbol{q}', i\nu') ,
\end{aligned}
$$
(C.10)

where $\int \equiv \int \frac{d^2\mathbf{k} d^2\mathbf{q}' d\omega d\nu'}{(2\pi)^6}$ and summation over the patch index $\theta$ is implied. By a simple change of frequency variables, we see that these two terms partially cancel, leaving us with

$$\Pi^{(\text{SE},1)} + \Pi^{(\text{SE},2)} = -\frac{g(\theta)^4}{i\nu} \int G_\theta \left( \mathbf{k} - \frac{\mathbf{q}'}{2}, i\omega \right) G_\theta \left( \mathbf{k} - \frac{\mathbf{q}'}{2}, i\omega + i\nu \right)$$
$$\cdot \left[ G_\theta \left( \mathbf{k} + \frac{\mathbf{q}'}{2}, i\omega + i\nu + i\nu' \right) - G_\theta \left( \mathbf{k} + \frac{\mathbf{q}'}{2}, i\omega + i\nu' \right) \right] D(\mathbf{q}', i\nu') . \tag{C.11}$$

The vertex correction diagram can be massaged into the same form using (C.7)

$$\Pi^{(\text{MT})} = -g(\theta)^4 \int G_\theta \left( \mathbf{k} - \frac{\mathbf{q}'}{2}, i\omega \right) G_\theta \left( \mathbf{k} - \frac{\mathbf{q}'}{2}, i\omega + i\nu \right)$$
$$\cdot G_\theta \left( \mathbf{k} + \frac{\mathbf{q}'}{2}, i\omega + i\nu' \right) G_\theta \left( \mathbf{k} + \frac{\mathbf{q}'}{2}, i\omega + i\nu + i\nu' \right) D(\mathbf{q}', i\nu')$$
$$= \frac{g(\theta)^4}{i\nu} \int G_\theta \left( \mathbf{k} - \frac{\mathbf{q}'}{2}, i\omega \right) G_\theta \left( \mathbf{k} - \frac{\mathbf{q}'}{2}, i\omega + i\nu \right)$$
$$\cdot \left[ G_\theta \left( \mathbf{k} + \frac{\mathbf{q}'}{2}, i\omega + i\nu' + i\nu \right) - G_\theta \left( \mathbf{k} + \frac{\mathbf{q}'}{2}, i\omega + i\nu' \right) \right] D(\mathbf{q}', i\nu') . \tag{C.12}$$

Comparing the expressions above, we see that

$$\Pi^{(\text{SE},1)} + \Pi^{(\text{SE},2)} + \Pi^{(\text{MT})} = 0 . \tag{C.13}$$

## C.3  Cancellation of Aslamasov-Larkin diagrams

The AL diagrams contain two fermion loops and hence a factor of $(-1)^2$. In terms of a loop integral

$$I_\theta(\mathbf{q}', \nu', \nu) = \int_{\mathbf{k}, \omega} G_\theta \left( \mathbf{k} + \frac{\mathbf{q}'}{2}, i\omega \right) G_\theta \left( \mathbf{k} + \frac{\mathbf{q}'}{2}, i\omega + i\nu \right) G_\theta \left( \mathbf{k} - \frac{\mathbf{q}'}{2}, i\omega - i\nu' \right) , \tag{C.14}$$

we can write the two diagrams as

$$\Pi^{(\text{AL},1)} = g(\theta)^3 g(\theta')^3 \int_{\mathbf{q}', \nu'} D(\mathbf{q}', i\nu') D(\mathbf{q}', i\nu' + i\nu) I_\theta(\mathbf{q}', \nu, \nu') I_{\theta'}(\mathbf{q}', \nu, \nu') , \tag{C.15}$$

$$\Pi^{(\text{AL},2)} = g(\theta)^3 g(\theta')^3 \int_{\mathbf{q}', \nu'} D(\mathbf{q}', i\nu') D(\mathbf{q}', i\nu' + i\nu) I_\theta(\mathbf{q}', \nu, \nu') I_{\theta'}(-\mathbf{q}', \nu, -\nu' - \nu) . \tag{C.16}$$

Adding up the two diagrams, we have

$$\Pi^{(\text{AL},1)} + \Pi^{(\text{AL},2)} \tag{C.17}$$

$$= g(\theta)^3 g(\theta')^3 \int_{\mathbf{q}', \nu'} D(\mathbf{q}', i\nu') D(\mathbf{q}', i\nu' + i\nu) I_\theta(\mathbf{q}', \nu, \nu') [I_{\theta'}(\mathbf{q}', \nu, \nu') + I_{\theta'}(-\mathbf{q}', \nu, -\nu' - \nu)] . \tag{C.18}$$

To make further progress, we recall the one-loop boson self energy at finite momentum and frequency

$$\Pi_\theta^{(0)}(\boldsymbol{q}', \nu') = g(\theta)^2 \int_{\boldsymbol{k}, \omega} G_\theta\left(\boldsymbol{k} + \frac{\boldsymbol{q}'}{2}, i\omega\right) G_\theta\left(\boldsymbol{k} - \frac{\boldsymbol{q}'}{2}, i\omega - i\nu'\right). \tag{C.19}$$

By a shift of frequency integration variables, we can easily show that

$$\Pi_\theta^{(0)}(-\boldsymbol{q}', -\nu') = g(\theta)^2 \int_{\boldsymbol{k}, \omega} G_\theta\left(\boldsymbol{k} - \frac{\boldsymbol{q}'}{2}, i\omega\right) G_\theta\left(\boldsymbol{k} + \frac{\boldsymbol{q}'}{2}, i\omega + i\nu'\right) = \Pi_\theta^{(0)}(\boldsymbol{q}', \nu'). \tag{C.20}$$

In terms of $\Pi^{(0)}$,

$$\begin{aligned}
&[I_{\theta'}(\boldsymbol{q}', \nu, \nu') + I_{\theta'}(-\boldsymbol{q}', \nu, -\nu' - \nu)] \\
&= \frac{1}{i\nu}\left[\Pi_{\theta'}^{(0)}(\boldsymbol{q}', \nu') - \Pi_{\theta'}^{(0)}(\boldsymbol{q}', \nu' + \nu) + \Pi_{\theta'}^{(0)}(-\boldsymbol{q}', -\nu' - \nu) - \Pi_{\theta'}^{(0)}(-\boldsymbol{q}', -\nu')\right] \\
&= \frac{1}{i\nu}\left[\Pi_{\theta'}^{(0)}(\boldsymbol{q}', \nu') - \Pi_{\theta'}^{(0)}(\boldsymbol{q}', \nu') + \Pi_{\theta'}^{(0)}(\boldsymbol{q}', \nu' + \nu) - \Pi_{\theta'}^{(0)}(\boldsymbol{q}', \nu' + \nu)\right] \\
&= 0
\end{aligned} \tag{C.21}$$

Since the above term vanishes for any choice of patch $\theta'$, we have demonstrated the vanishing of $\Pi^{(\mathrm{AL},1)} + \Pi^{(\mathrm{AL},2)}$ independent of the choice of form factor $g(\theta)$.

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
