# Peer review of "Gifts from anomalies: Exact results for Landau phase transitions in metals"

_SciPost Physics_

## Round 2 · Referee Report · Luca Delacrétaz (Referee 1) · 2022-7-15

Report

This manuscript proposes to find nonperturbative results in a large class of non-Fermi liquids (NFLs) by using the emergent anomalous symmetries of a "mid-IR" effective theory that is expected to flow to such phases. A central tool is an anomaly similar (but not identical) to the one considered in previous work by some of the authors [41]. While anomalies are commonly used to constrain gapped phases of matter, and the infrared of particle physics-like QFTs, little work has attempted to make use of them in the much more gapless world of Fermi liquids. This paper therefore provides a valuable new approach to the difficult problem of taming NFLs.

The main result of this work is to obtain on general grounds bosonic correlation functions at zero wavevector $q=0$, namely the propagator of the boson (which can be an emergent gauge field or a critical order parameter), and the optical conductivity. Since $q=0$ observables are away from the quantum critical scaling regime $\omega\sim q^z \ll q$ ($z\sim 3$), it is perhaps not entirely surprising that they are fixed without the need to solve for the dynamics. Nevertheless, these results are by no means obvious and, as the authors point out, are in fact not automatically reproduced by certain approximations in the literature. Moreover, these observables are experimentally relevant. So these results, if correct, are welcome.

The manuscript is thorough, and discusses comparison with much of the existing literature. However, a few technical aspects deserve clarifications before publication:

  • Below Eq. (3.25), the authors argue that the current operator is effectively given by its component perpendicular to the Fermi surface (3.22), up to irrelevant corrections. This seems to imply that within the same approximation, the U(1)_patch current $j_\theta$ is proportional to $n_\theta$, and therefore that the anomaly equation (3.12) only involves the operators $n_\theta$ and $\phi$, allowing to close the equations in Sec. 4 and generalize these results to finite $q\neq 0$. This seems like a (possibly too) strong conclusion. Could the authors explain why they believe it is invalid?

  • Operator relations such as the ones used throughout this paper usually hold up to contact terms, giving possible analytic contributions to finite frequency correlators. Given that the $q=0$ boson correlator that the authors find is analytic, one might worry that while a non-analytic dependence on $\omega$ at the fixed point is ruled out, the coefficient of the analytic (constant) piece is not unambiguously fixed by this approach. Can this be ruled out?

  • Could the authors comment on how their approach (in particular their anomalous Ward identity for emergent symmetries) relates to or differs from "asymptotic Ward identities" sometimes discussed in the same context? See, e.g., [Metzner Castellani and Di Castro, "Fermi systems with strong forward scattering.", Advances in Physics 47.3 (1998): 317-445.] and references therein.

  • Finally, can the cancellations of diagrams in appendix C be understood simply by the fact that they all involve a correlation function of a total charge $Q_\theta = n_\theta(q=0)$ in the unperturbed Fermi liquid patch theory, which is a trivial operator? (specifically, each diagram involves a fermion loop with a $q=0$ external leg). This may offer a perturbative explanation for the RPA-exactness of the author's result.

I recommend this manuscript for publication once the points above are addressed.

  • validity: high
  • significance: top
  • originality: top
  • clarity: high
  • formatting: perfect
  • grammar: perfect

Author:  Zhengyan Shi  on 2022-08-26  [id 2757]

(in reply to Report 1 by Luca Delacrétaz on 2022-07-15)

Referee's point 1:

Below Eq. (3.25), the authors argue that the current operator is effectively given by its component perpendicular to the Fermi surface (3.22), up to irrelevant corrections. This seems to imply that within the same approximation, the $\mathrm{U}$(1)patch current $j_{\theta}$ is proportional to $n_{\theta}$, and therefore that the anomaly equation (3.12) only involves the operators $n_{\theta}$ and $\phi$, allowing to close the equations in Sec. 4 and generalize these results to finite $\mathbf{q} \neq 0$. This seems like a (possibly too) strong conclusion. Could the authors explain why they believe it is invalid?

Response: Indeed, our anomaly arguments are not strong enough to fix the boson correlators at $\mathbf{q} \neq 0$. This is because the U$(1)$patch anomaly equation involves both the components of the current parallel and perpendicular to the Fermi surface (FS),

$$ \partial_t n_{\theta}+\mathbf{\nabla}\cdot\mathbf{j}\theta= \partial_t n} + \nabla_{\perp}\, j^{\perp{\theta} + \nabla}\, j^{||{\theta} = - \frac{\Lambda\, g_a(\theta)\, \partial_t \phi^a \,. $$}}{(2\pi)^2\, v_F(\theta)

As the referee points out, at the IR fixed point, one might expect the term involving $j^{||}(\theta) = v_F(\theta)\, \psi^{\dagger}(\theta) \nabla_{||} \psi(\theta)$ to be unimportant at low energies compared to $j^{\perp}(\theta) = v_F(\theta) \psi^{\dagger}(\theta) \psi(\theta)$, such that the anomaly equation can be reduced to a closed equation involving $n_\theta$ and $\phi$ alone. However, the patch curvature (and thus $j^{\parallel}(\theta)$) should always be included in any ``mid-IR'' patch theory to guarantee the correct low energy behavior. This is consistent with the commonly used patch scaling procedure, where $\nabla_{\perp} \sim \nabla^2_{||}$, and thus $\nabla_{\perp} j^{\perp}(\theta)$ and $\nabla_{||} j^{||}(\theta)$ have the same tree level scaling. The only consistent way to drop the curvature of the dispersion completely is to include large-angle scattering between patches, which here we are seeking to neglect. Therefore, we cannot write down a closed set of equations for $n_{\theta}, \phi$ and solve for the $\mathbf{q} \neq 0$ correlators.

Regarding the dropping of $j_\theta^{\parallel}$ in calculating the optical conductivity, we emphasize that this argument required an assumption of emergent intra-patch momentum conservation occuring at a scale below that of the mid-IR theory (which is justified by earlier perturbative approaches to this problem, see e.g. [Metlitski and Sachdev, PRB (2010)]). Because this then implies conservation of $j_\theta^\parallel$, the only contributions to the optical conductivity will come from $j^\perp_\theta$, which is not conserved. Importantly, none of this is equivalent to a statement of irrelevance of the curvature of the FS at the level of the mid-IR theory.

Referee's point 2:

Operator relations such as the ones used throughout this paper usually hold up to contact terms, giving possible analytic contributions to finite frequency correlators. Given that the $\mathbf{q} = 0$ boson correlator that the authors find is analytic, one might worry that while a non-analytic dependence on $\omega$ at the fixed point is ruled out, the coefficient of the analytic (constant) piece is not unambiguously fixed by this approach. Can this be ruled out?

Response: We can make two comments regarding contact terms. First, we note that contact terms indeed play an important role in our arguments. For example, as discussed in Section 3.5, the conductivity $\sigma(\mathbf{q} = 0,\omega)$ receives an important contribution from a contact term for the external electromagnetic field. Without this contact term, if we were to directly use the anomaly equation to make the operator substitution,

$$ J^i_{\theta}(\mathbf{q}=0) \rightarrow -\frac{\Lambda_{\rm patch}}{(2\pi)^2} w^i(\theta) g_a(\theta)\, \phi^a(\mathbf{q}=0) \,, $$
we would conclude that $\sigma(\mathbf{q}=0,\omega)$ is proportional to the boson correlator $D(\mathbf{q}=0,\omega)$, which would imply a vanishing conductivity for e.g. an Ising-nematic order parameter (which of course is not correct). However, as the referee rightly points out, this conclusion misses an important analytic contribution, essentially corresponding to a diamagnetic term in the current. In equations (3.27) through (3.32), we explain how we keep track of contact terms in the path integral approach and ultimately obtain (3.35) which should be free from additional analytic corrections.

Nevertheless, one may worry that there are additional contact terms involving higher powers of the external EM probes that we have missed. While we cannot prove that no such terms need to be included in the analysis of our mid-IR theory, the Fujikawa-style derivation of the anomaly equation in Appendix B suggests that any such terms vanish, at least in the low energy limit chosen there. We note also that, if such terms were nonvanishing, then presumably there would also be terms involving higher powers of the boson field (since it also probes a subgroup of U$(1)_{\mathrm{patch}}$ that is not anomalous), which would invalidate the basic operator equivalence.

Referee's point 3:

Could the authors comment on how their approach (in particular their anomalous Ward identity for emergent symmetries) relates to or differs from "asymptotic Ward identities" sometimes discussed in the same context? See, e.g., [Metzner Castellani and Di Castro, "Fermi systems with strong forward scattering.", Advances in Physics 47.3 (1998): 317-445.] and references therein.

Response: There is certainly a relationship between the two, as the anomaly equation used throughout our work can be understood as generating a family of anomalous Ward identities. However, within the patch construction, the asymptotic Ward identities reviewed in [Metzner, Castellani, and Di Castro, Adv. Phys. (1998)] require neglecting the dispersion parallel to the FS, or equivalently the FS curvature. But, as mentioned in response to the referee's first question, perturbative calculations of response functions in models of this type have shown that the curvature of the FS must be included to account for the leading small angle scattering processes (this point is also explained on p. 112 of Metzner, Castellani, and Di Castro). The novel point in our work is that even when the parallel dispersion is included, a useful anomaly equation can be derived, although it doesn't fix the boson correlator at $\mathbf{q} \neq 0$.

Referee's point 4:

Finally, can the cancellations of diagrams in appendix C be understood simply by the fact that they all involve a correlation function of a total charge $Q_{\theta} = n_{\theta}(\mathbf{q} = 0)$ in the unperturbed Fermi liquid patch theory, which is a trivial operator? (specifically, each diagram involves a fermion loop with a $\mathbf{q} = 0$ external leg). This may offer a perturbative explanation for the RPA-exactness of the author's result.

Response: Because the anomaly indeed fixes $n_\theta(\omega,\mathbf{q}=0)$ to be proportional to the boson field operator, $g^a(\theta)\,\phi_a(\omega,\mathbf{q}=0)$, the boson self-energy can be seen to be one-loop exact using the arguments in Section 3. One might therefore think of the calculation performed in Appendix C as a perturbative justification of this conclusion.

---

## Round 2 · Referee Report · Ilya Esterlis (Referee 2) · 2022-7-20

Strengths

  1. Highly original work
  2. Clear presentation
  3. Scholarly account of connections to existing literature

Report

In "Gifts from anomalies: Exact results for Landau phase transitions in metals," the authors demonstrate how a combination of renormalization group and anomaly arguments constrain the low-energy behavior of an important class of physical problems, that of a Fermi surface coupled to a fluctuating Landau order parameter. Such problems are of broad importance for a variety of interesting materials, yet have also been notoriously challenging to address by conventional theoretical means. The authors bring a new perspective and approach to the problem, and in doing so derive a number of new and interesting results.

The central results are a non-perturbative calculation of the propagator of the order parameter field in the somewhat unusual regime of $q=0$ and finite $\omega$, as well as a related calculation of the optical conductivity. The latter calculation shows that, in the deep IR limit, the optical conductivity of such a system consists of a single Drude peak. That is, they demonstrate the absence of an incoherent contribution to the conductivity with $\omega/T$ scaling, which is believed to be characteristic of the quantum critical regime. This result is especially interesting and important, given that the optical conductivity is a standard experimental probe.

I also found the presentation to be particularly lucid and clear. As a non-expert on some the anomaly-type arguments, I was appreciative of Section 2, which provided simple examples and analogies. Furthermore, the thorough comparisons with earlier work in Section 5 were very helpful in placing the results in context and understanding the differences between various approaches.

I have a few small suggestions/questions that I hope may strengthen the paper:

  1. There has been a significant amount of insight gained from Quantum Monte Carlo studies of the metallic Ising-nematic quantum critical point. In Y. Schattner, et. al., "Ising Nematic Quantum Critical Point in a Metal: A Monte Carlo Study" PRX 6, 031028 (2016) the authors describe various normal state properties of the system. I wonder if there are connections between what was found in that study and what the current authors have found? For instance, the inset of Fig.12 shows precisely the boson propagator at $q=0$ and finite omega. Neglecting the point exactly at $\omega=0$, the behavior seems to be compatible with the findings of the present paper; i.e. $D(q=0,\omega)$ tends to a constant at small frequency. In followup work [Lederer, et. al., "Superconductivity and non-Fermi liquid behavior near a nematic quantum critical point", PNAS 114, 4905 (2017)] the same authors described the behavior of the optical conductivity in the system. It might be interesting to revisit their data with the results of the current work in mind.

  2. In Sec. 3.6, my understanding is that some of the results are actually the same as properties of the simple non-interacting particle-hole bubble, and hence have a simple perturbative interpretation. For instance, in the regularization used by the authors, Eq. 3.45 would also follow from the vanishing of the particle-hole bubble at omega=0 in perturbation theory. By the same token, the subsequent discussion of the vanishing compressibility is also a problem if ones compute the non-interacting particle-hole bubble with this regularization. Of course, the virtue of the present work is the power to make non-perturbative statements, but it may help orient the reader to provide these simple analogies (assuming I am correct).

  3. Before Eq. 4.5, the authors mention "standard caveats around such statements." While the subsequent discussion is fairly clear, I think it would be useful to provide a reference here for the reader interested in delving further into these caveats, in particular for the "secondary" fixed-point scaling.

  4. Below Eq. 6.8 , the authors write "...and the critical point at $r=0$..." -- perhaps I missed it, but I could not find where the parameter $r$ had been defined.

Overall, I believe this is high-quality work that provides a welcome new perspective in the field of metallic quantum criticality, and I recommend it for publication.

  • validity: top
  • significance: top
  • originality: top
  • clarity: high
  • formatting: perfect
  • grammar: perfect

Author:  Zhengyan Shi  on 2022-08-26  [id 2756]

(in reply to Report 2 by Ilya Esterlis on 2022-07-20)

  1. The referee's point about the two recent DQMC studies is a good one, and we have added comments to the Discussion comparing our conclusions about $\sigma(\mathbf{q} = 0, \omega)$ and $D(\mathbf{q}=0,\omega)$ to the DQMC results (adding references as well). As the referee observes, the saturation of $\Pi(\mathbf{q}=0,\omega) = D^{-1}(\mathbf{q}=0,\omega)$ to a constant value at $\omega = 0$ is consistent with our results. The residual frequency dependence in $\Pi(\mathbf{q}=0,\omega)$ found in the DQMC study likely comes from irrelevant operators included in the lattice model that are outside the scope of our analysis (see also the discussion in Section 4 of the manuscript). As for the optical conductivity $\sigma(\mathbf{q}=0,\omega)$, the DQMC studies find a finite-width Drude peak at low frequency, which one expects due to the presence of irrelevant operators in the lattice model that degrade momentum (e.g. Umklapp processes). A precise characterization of these irrelevant operators would be needed to match the DQMC results in detail. Finally, we note that all the DQMC results are technically obtained at Matsubara frequencies rather than real frequencies. The extraction of $\Pi(\mathbf{q}=0, \omega)$ and $\sigma(\mathbf{q}=0,\omega)$ requires a subtle numerical analytic continuation with errors that are not systematically controlled.

  2. Indeed, the content of the anomaly relation can be interpreted as the statement that $\Pi(\mathbf{q}=0,\omega)$ is one-loop exact (see also the response to the fourth point brought up by Referee 1), up to the presence of suitable regularization-dependent contact terms that the referee alludes to.

  3. We have added a reference to relevant chapters of the classic lecture notes [Goldenfeld, CRC Press, 2018] that contextualize our discussion. Admittedly, ``secondary fixed point scaling'' is actually terminology of our own invention. To our knowledge, it has not appeared in the literature before, and the concept is not often presented in juxtaposition with the standard concept of corrections to scaling (as we do in this manuscript).

  4. We thank the referee for pointing out this mistake. $r$ is the parameter we tune to access the QCP. Elsewhere in the draft, this parameter is denoted $m_c^2$. We have removed the $r$ notation from the manuscript.

---

## Round 2 · Referee Report · Anonymous (Referee 3) · 2022-7-23

Strengths

  1. The main result is very useful, argument is novel and seems solid.

  2. Presentation is very clear.

  3. Comparison with previous literatures is especially helpful.

Report

This paper obtained some (surprising) exact results of the dynamics of bosonic fields coupled with systems with a Fermi surface, using the 't Hooft anomaly of the large emergent symmetries at the infrared limit of the system. The method used in this paper is completely nonperturbative, and the exact result is different from previous literatures which used perturbative diagrammatic methods.

In general I really liked this paper, and I find the result very useful. The analysis of anomaly is a new method introduced by the authors (in this paper and their previous papers), and now this method seems rather powerful. The authors also made detailed comparison with previous works which used traditional diagrammatic perturbative methods. This comparison makes the work rather complete.

I just have one suggestion for the authors to consider, and it is optional.

I am curious under what condition would the argument for the exact result fail. When would the boson acquire a self-energy that is not a constant in the limit $q-->0$, even when the boson is coupled to a system with a finite Fermi surface? We can still use the 1d fermion system as a starting point. Part of the argument in the paper uses the fact that the "chiral density" is the "charge current"; but this relation seems to hold only when the velocity of the fermion is finite at the Fermi surface. One can think of scenarios where the Fermi velocity is singular near the Fermi surface, which seems to invalidate the relation between the chiral density and charge current (as now the charge current is no longer just $\psi^\dagger_+ \psi_+ - \psi^\dagger_- \psi_-$, but more like $ k^\eta (\psi^\dagger_+ \psi_+ - \psi^\dagger_- \psi_-)$, where the exponent $\eta$ is nonzero due to the singularity of Fermi velocity). Singular Fermi velocity corresponds to singularity of density of states (some kind of Van Hove singularity) in higher dimensions. Does this mean that when there is VHS at the Fermi surface, the bosons can acquire more nontrivial self-energy?

It would be good to clarify this point, but as I said, it is optional.
  • validity: high
  • significance: high
  • originality: high
  • clarity: high
  • formatting: excellent
  • grammar: excellent

Author:  Zhengyan Shi  on 2022-08-26  [id 2755]

(in reply to Report 3 on 2022-07-23)

The referee brings up what is perhaps the simplest situation in which our arguments would fail; namely, a Fermi surface with Van Hove points. Although the anomaly constraints we derive should be valid in regions where the Fermi surface is regular, they will break down as one approaches the vicinity of the Van Hove singularity. It is an interesting future direction to understand what non-perturbative statements can be made about transport in this context, which may be pertinent to some interesting material contexts.

Other situations where our arguments are affected involve systems where the $\rm{U}(1)_{\mathrm{patch}}$ symmetry is broken explicitly and is not restored in the IR. The most innocuous such example is a system with an inversion-odd order parameter (which mediates a repulsive interaction) in the presence of BCS pairing. Such systems flow to an IR fixed point with finite BCS couplings, breaking the $\rm{U}(1)_{\mathrm{patch}}$ symmetry down to a subgroup. We discuss this case in Section 3.7 of the manuscript. More serious damage is done by quenched disorder, which generically breaks the $\rm{U}(1)_{\mathrm{patch}}$ symmetry completely by coupling all of the patches of our mid-IR theory.

---

## Editorial Decision

resubmitted